# Disappearing cities on US coasts

Leonard O. Ohenhen[1,2 ✉], Manoochehr Shirzaei[1,2,3], Chandrakanta Ojha[4], Sonam F. Sherpa[5,6] & Robert J. Nicholls[7]

The sea level along the US coastlines is projected to rise by 0.25–0.3 m by 2050, increasing the probability of more destructive flooding and inundation in major cities[1–3]. However, these impacts may be exacerbated by coastal subsidence— the sinking of coastal land areas[4]—a factor that is often underrepresented in coastal-management policies and long-term urban planning[2,5]. In this study, we combine high-resolution vertical land motion (that is, raising or lowering of land) and elevation datasets with projections of sea-level rise to quantify the potential inundated areas in 32 major US coastal cities. Here we show that, even when considering the current coastal-defence structures, further land area of between 1,006 and 1,389 km² is threatened by relative sea-level rise by 2050, posing a threat to a population of 55,000–273,000 people and 31,000–171,000 properties. Our analysis shows that not accounting for spatially variable land subsidence within the cities may lead to inaccurate projections of expected exposure. These potential consequences show the scale of the adaptation challenge, which is not appreciated in most US coastal cities.

The widespread consequences of global climate change stress human communities and ecosystems worldwide. Climate change is already causing an increase in the frequency and intensity of heatwaves, hurricanes and wildfires and severely affecting the world's freshwater resources through sea-level rise (SLR), more frequent droughts and changes in precipitation and evapotranspiration[6–8], and these effects will almost certainly grow in the future[9]. Globally, SLR will pose a substantial socioeconomic challenge in the twenty-first century, primarily affecting human populations, infrastructure and ecosystems along major coastlines[10–14]. Global mean sea level has risen by about 0.17 m over the past 100 years, with global rates of SLR accelerating from about 1.7 mm per year in the late twentieth century to about 3.1 mm per year in the early twenty-first century in response to warming temperatures[3,15–17] and is 3.7 mm per year at present (ref. 17). Even if climate change mitigation efforts succeed in stabilizing temperature in the future decades, sea levels will continue to rise as a result of the continuing response of oceans to past warming[3,17,18]. Furthermore, coastal cities often experience sinking land (so-called land subsidence), whose compounding effect contributes to relative SLR, exacerbating coastal hazards and risks[2,5,19,20]. In this study, we refer to SLR that incorporates the effects of vertical land motion (VLM) as 'relative SLR', whereas 'geocentric SLR' refers to SLR without VLM. As sea level rises and land subsides, the hazards associated with climate extremes (for example, hurricanes and storm surges), shoreline erosion and inundation of low-lying coastal areas grow[21,22].

On the coasts of the conterminous USA, climate-induced sea levels are rising faster than the global average[3], with an expected increase over the next few decades (Fig. 1 and Extended Data Fig. 1). Owing to its geography and population distribution, the USA is a coastal nation, with more than 30% of its population residing in coastal cities, generating an estimated annual revenue of US$3.8 trillion (ref. 23).

Consequently, socioeconomic losses from climate-induced SLR will represent a notable facet of climate-change consequences in the USA[24]. In the short term (one to three decades), only continued observed rises in sea level are sufficient to trigger cascading hazards across US coastal regions, with a projected increase in the frequency and intensity of storm surges, saltwater intrusion, high-tide flooding and coastal erosion[3,12,16,25,26].

Sea levels are projected to vary minimally in the next few decades across the different greenhouse gas emission scenarios, whereas end-of-century projections indicate a more substantial divergence in increase by emissions (Extended Data Fig. 1b–g). Thus, short-term vulnerability assessments incorporating local drivers are relevant to policymakers for developing comprehensive adaptation strategies that extend beyond emission reduction, as they provide insights into the immediate risks and challenges of coastal communities. However, accurately projecting coastal vulnerability requires comprehensive and high-resolution measurements of VLM, which is lacking across the USA at present[2]. This lack of data makes estimating the actual risks of relative SLR on different coastal communities challenging and contributes to high uncertainty and potentially systematic errors in existing coastal-hazard assessment.

Here we present coastal-scenario-based inundation hazard models for coastal cities in the USA, focusing on the short-term (2050) projection of relative SLR. Our inundation models integrate high-resolution VLM data across the US coast[27,28] (Fig. 1 and Extended Data Figs. 2–4) at millimetre-level accuracy using interferometric synthetic aperture radar (InSAR) (see Methods), the projections of geocentric SLR and light detection and ranging (LiDAR) digital elevation models (DEMs) to forecast relative SLR rates and create granular inundation maps for a set of major cities along the US coastline. Using the 2010 US census data as baseline estimates, we assess the impact of relative SLR on the

[1]Department of Geosciences, Virginia Tech, Blacksburg, VA, USA. [2]Virginia Tech National Security Institute, Virginia Tech, Blacksburg, VA, USA. [3]Institute for Water, Environment and Health, United Nations University, Hamilton, Ontario, Canada. [4]Department of Earth and Environmental Sciences, IISER Mohali, Punjab, India. [5]Department of Earth, Environmental and Planetary Sciences, Brown University, Providence, RI, USA. [6]Institute at Brown for Environment and Society, Brown University, Providence, RI, USA. [7]Tyndall Centre for Climate Change Research, University of East Anglia, Norwich, UK. ✉e-mail: ohleonard@vt.edu

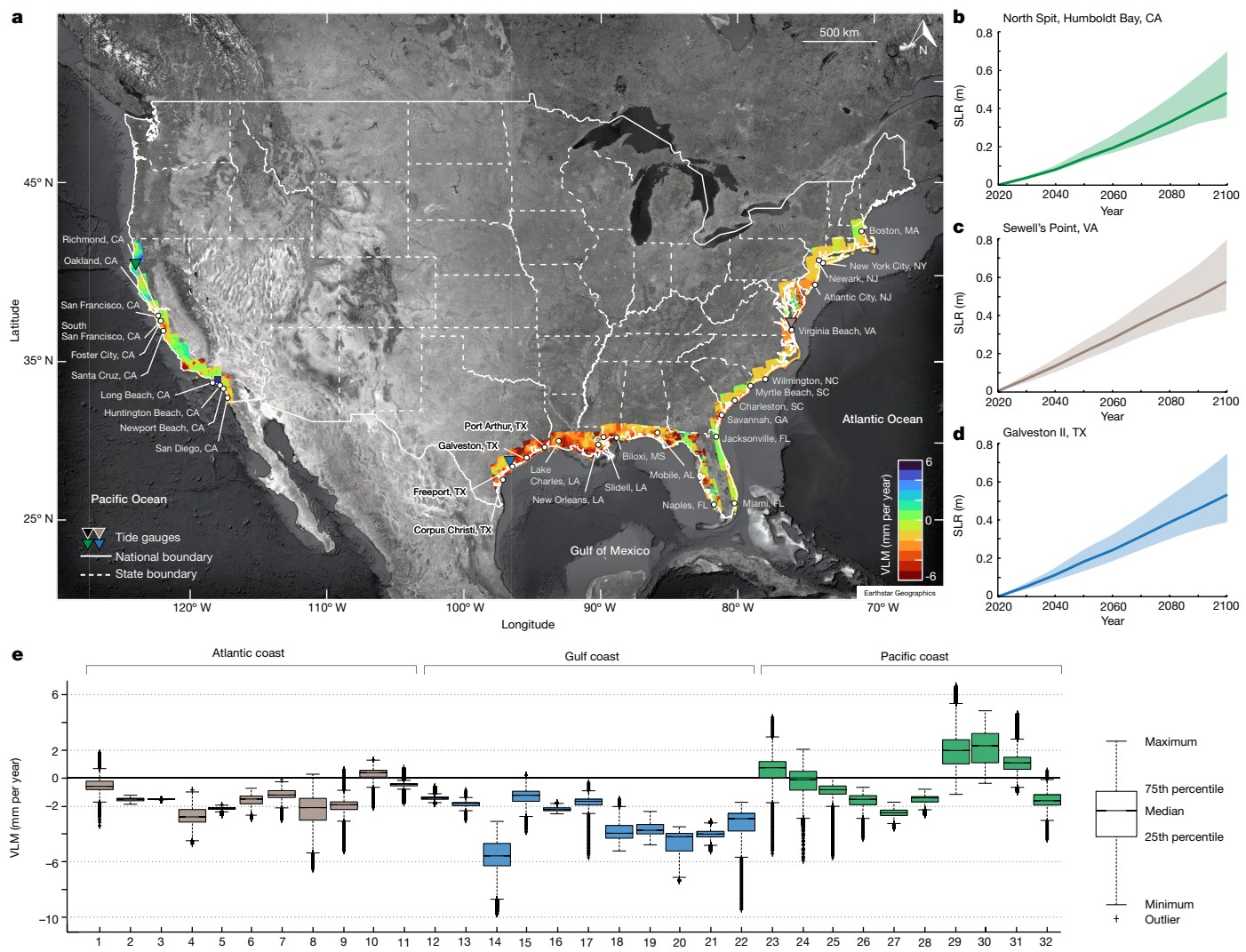

**Fig. 1 | Coastal hazards across the USA. a**, Spatial distribution of VLM across the US coast (background image: Google, Earthstar). Positive VLM rates indicate uplift and negative VLM rates indicate subsidence. Individual VLM maps for the US Atlantic, Gulf and Pacific coasts are shown in Extended Data Figs. 2–4. National and state boundaries in **a** are based on public-domain vector data by the World Bank DataBank (https://data.worldbank.org/) generated in MATLAB. Projections of geocentric SLR with a baseline for the year 2020 based on SSP2-4.5 (refs. 17,34) on the Pacific coast (North Spit, Humboldt Bay, CA) (**b**), Atlantic coast (Sewell's Point, VA) (**c**) and Gulf coast (Galveston II, TX) (**d**). The bold lines represent the median (50th percentile) projected geocentric SLR and the shaded regions represent the 17th and 83rd percentiles. **e**, Box plot representing the distribution of VLM for 32 US coastal cities evaluated in this study. The 32 coastal cities evaluated in this study are highlighted in **a**. The cities include: US Atlantic coast: 1. Boston, MA; 2. New York City, NY; 3. Jersey City, NJ; 4. Atlantic City, NJ; 5. Virginia Beach, VA; 6. Wilmington, NC; 7. Myrtle Beach, SC; 8. Charleston, SC; 9. Savannah, GA; 10. Jacksonville, FL; 11. Miami, FL; US Gulf coast: 12. Naples, FL; 13. Mobile, AL; 14. Biloxi, MS; 15. New Orleans, LA; 16. Slidell, LA; 17. Lake Charles, LA; 18. Port Arthur, TX; 19. Texas City, TX; 20. Galveston, TX; 21. Freeport, TX; 22. Corpus Christi, TX; US Pacific coast: 23. Richmond, CA; 24. Oakland, CA; 25. San Francisco, CA; 26. South San Francisco, CA; 27. Foster City, CA; 28. Santa Cruz, CA; 29. Long Beach, CA; 30. Huntington Beach, CA; 31. Newport Beach, CA; 32. San Diego, CA.

population and properties of US coastal communities and explore several adaptation regimes to minimize potential future impacts.

## Future socioeconomic impact of relative SLR on US coasts

Future inundation of coastal areas in 32 coastal cities (see Fig. 1) along the US coasts is modelled using projections of current VLM rates on coastal-elevation data, geocentric SLR projection scenarios and high-tide estimates. The aggregate population of these cities is estimated to be 25 million people (roughly 20% of current US coastal inhabitants), with 10 million properties valued at US$12 trillion (Supplementary Table 1). Our analysis quantifies how relative sea-level changes, attributable to VLM and geocentric SLR, will increase the exposure—area, population and properties—to high-tide flooding by 2050, using 2020 as the baseline (see Methods). For the analysis, we consider the SLR scenario derived from Shared Socioeconomic Pathway (SSP) scenario 2-4.5 (SSP2-4.5), representing the current emissions trajectory[9], population and property exposure using the 2010 US census data and property/home value using the ZIP Code Zillow Home Value Index (ZHVI) (see Methods). Our model suggests that, by 2050, relative SLR could cause further exposed land area of between 1,334 and 1,813 km² in 32 US coastal cities if no flood-defence structures are implemented (Figs. 2–4, Table 1 and Supplementary Tables 2–4). We estimate a population exposure of 176,000–518,000 inhabitants from 94,000–288,000 properties,

Atlantic coast (year 2050: SSP2-4.5)

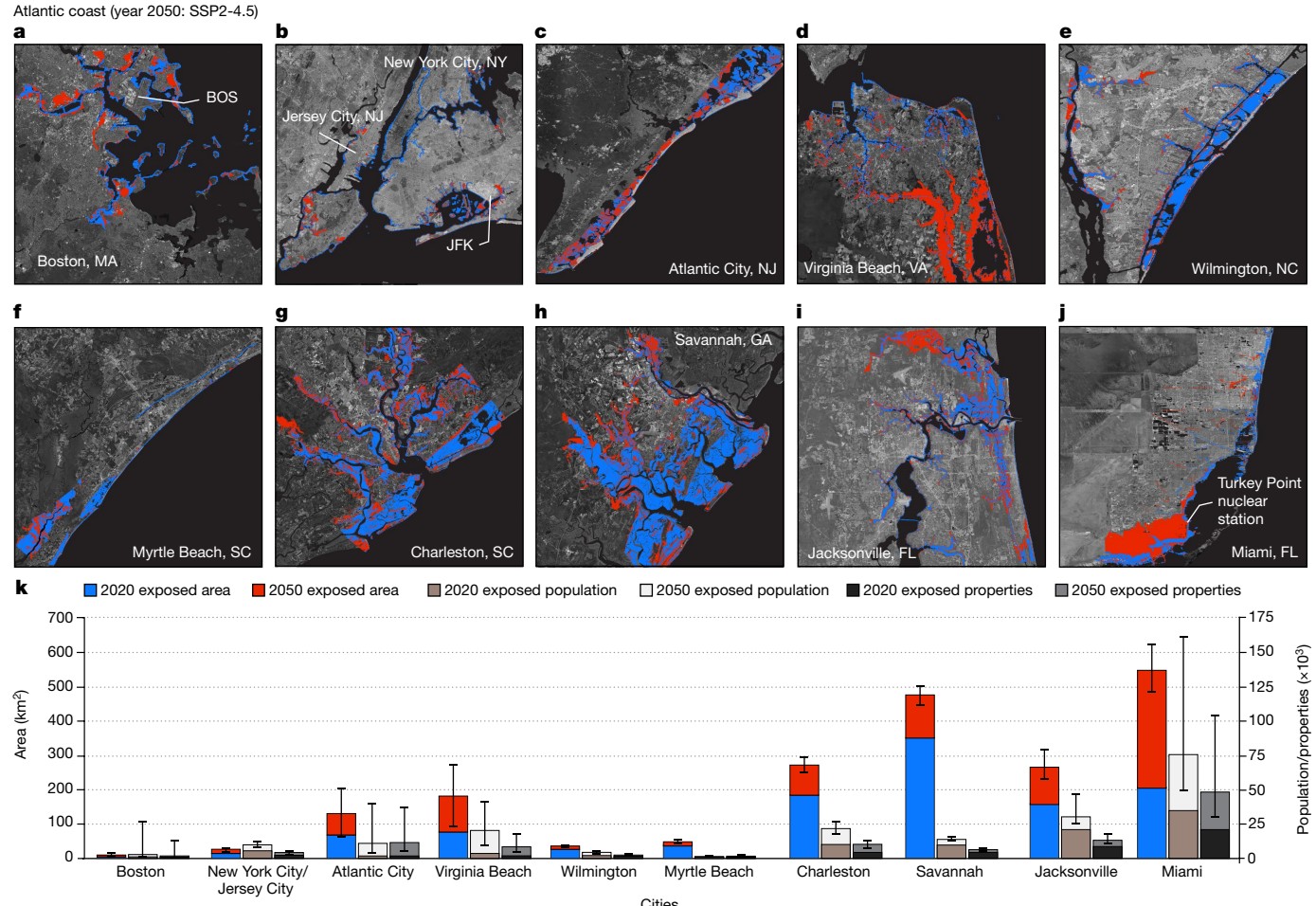

**Fig. 2 | Inundation maps for cities on the US Atlantic coast.** Areas exposed to current (2020) high tide and further exposed areas by 2050 considering VLM, geocentric SLR projection under SSP2-4.5 for: Boston, MA (**a**); New York City, NY and Jersey City, NJ (**b**); Atlantic City, NJ (**c**); Virginia Beach, VA (**d**); Wilmington, NC (**e**); Myrtle Beach, SC (**f**); Charleston, SC (**g**); Savannah, GA (**h**); Jacksonville, FL (**i**); and Miami, FL (**j**). Background images in **a**–**j** are from Google, Earthstar. Note that, for current exposure, geocentric SLR = 0 m. The blue and red colours are the current and projected exposed areas, respectively. BOS, Boston Logan International Airport, MA; JFK, John F. Kennedy International Airport, NY. **k**, Distribution of further exposed areas, population and properties by 2050. The central value represents the estimated median value from equation (3), whereas the error bars represent the lower and upper bounds from equation (3).

with a total estimated home value of US$32–109 billion by 2050. The maximum population and property exposure by 2050 represents approximately 1 in 50 people and 1 in 35 properties from the 32 coastal cities.

## Atlantic coast

From 11 cities on the US Atlantic coast, the projected extra area exposed to high-tide flooding by 2050 is between 773 and 951 km² (Fig. 2). This would affect a population of 59,000–263,000 people and 32,000–163,000 properties on the US Atlantic coast (Fig. 2 and Table 1). The property and population exposure on the Atlantic coast are not homogeneous across all cities. For example, Miami (average elevation of less than 2 m above sea level) has the greatest share of exposure, accounting for 38–44% (340–360 km²) of the exposed area, 38–46% (22,000–122,000) of the exposed population and 41–49% (13,000–81,000) of the exposed properties along the Atlantic coast (Supplementary Table 2). The home-value exposure by 2050 for the 11 Atlantic coastal cities is estimated at US$14–64 billion (Table 1). The calculated exposure does not account for the value of critical infrastructure (such as airports, schools, hospitals, power plants, roads and railways), as well as economic hubs and landmarks, and hence represents a conservative value.

## Gulf coast

For 11 cities along the US Gulf coast, our 2050 projection of inundation hazard shows a cumulative exposed area of between 528 and 826 km² (Fig. 3, Table 1 and Supplementary Table 3). These affected areas will expose an extra 110,000–225,000 people and 58,000–109,000 properties worth US$14–21 billion (Table 1). It should be noted that substantial areas (318–426 km²), population (386,000–448,000) and properties (176,000–209,000) in New Orleans are already exposed to high-tide events at present, owing to the existence of areas lying below sea level (Supplementary Table 3). We emphasize that our analysis applies an undefended approach, which does not consider the presence of flood-control structures and future adaptation. New Orleans, however, is surrounded by extensive floodwalls and levee systems and is heavily drained. Although flood-control structures offer substantial protection to coastal areas, their effectiveness is not guaranteed. The events following the landfall of Hurricane Katrina in August 2005, which claimed more than 1,500 lives, are tragic reminders of the heightened devastation of a failed levee system[29]. Thus, considerations of flood-control systems alone may only represent a temporary solution, as discussed later. Airports, roadways and refineries, ubiquitous in cities, are projected to be among the Gulf coast's exposed infrastructure (Fig. 3).

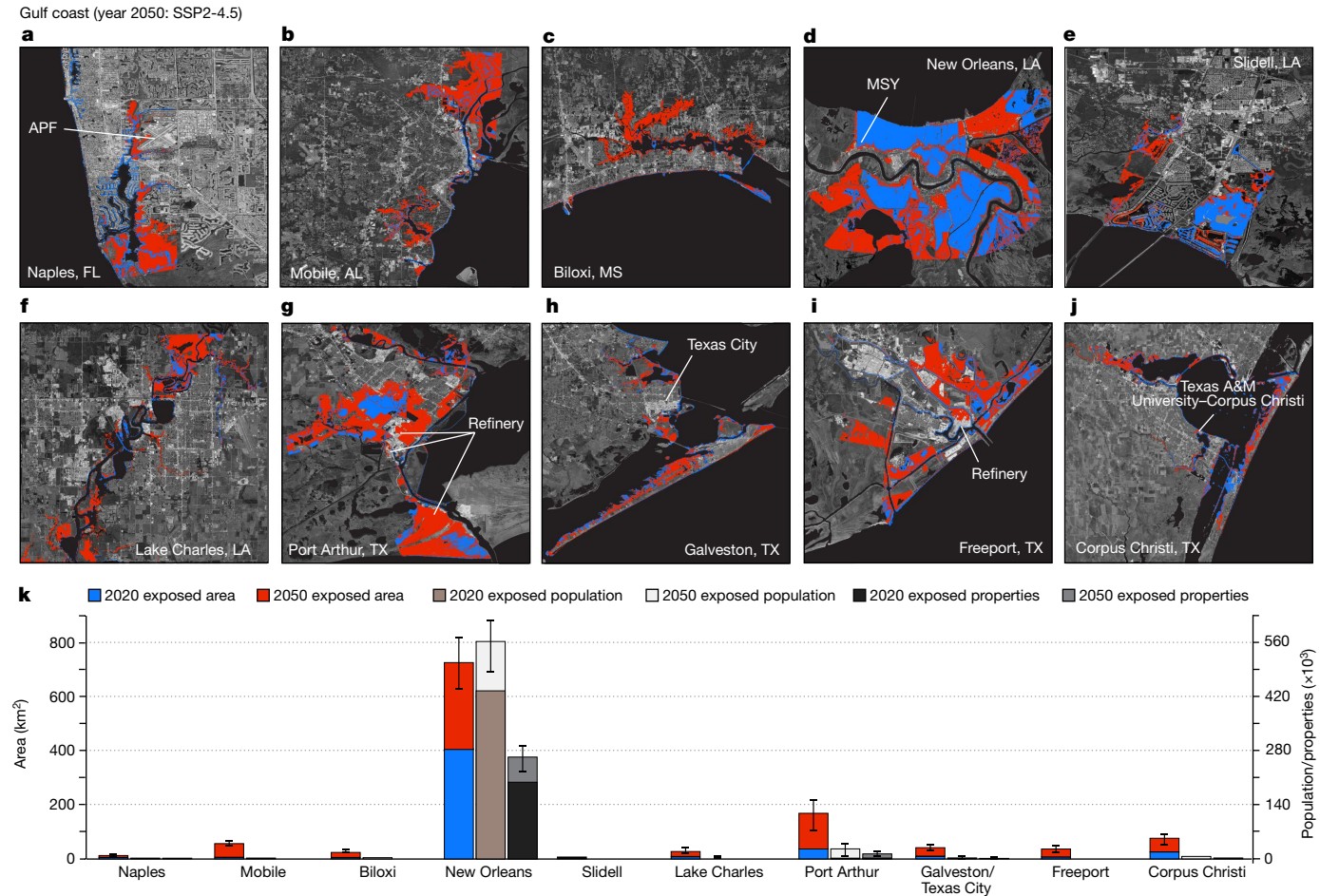

Gulf coast (year 2050: SSP2-4.5)

**Fig. 3 | Inundation maps for cities on the US Gulf coast.** Areas exposed to current (2020) high tide and further exposed areas by 2050 considering VLM, geocentric SLR projection under SSP2-4.5 for: Naples, FL (**a**); Mobile, AL (**b**); Biloxi, MS (**c**); New Orleans, LA (**d**); Slidell, LA (**e**); Lake Charles, LA (**f**); Port Arthur, TX (**g**); Galveston and Texas City, TX (**h**); Freeport, TX (**i**); and Corpus Christi, TX (**j**). Background images in **a**–**j** are from Google, Earthstar. Note that, for current exposure, geocentric SLR = 0 m. The blue and red colours are the current and projected exposed areas, respectively. APF, Naples Airport, FL; MSY, Louis Armstrong New Orleans International Airport, LA. **k**, Distribution of further exposed areas, population and properties by 2050. The central value represents the estimated median value from equation (3), whereas the error bars represent the lower and upper bounds from equation (3).

## Pacific coast

On the Pacific coast, we find a considerable divergence in the impacts of relative SLR compared with the US Atlantic and Gulf coasts. By 2050, the cumulative exposed area from ten cities on the US Pacific coast is 20–40 km$^2$, with a population exposure of 6,000–30,000 people and 3,000–15,000 properties worth US$4.5–22 billion (Fig. 4, Table 1 and Supplementary Table 4). The comparatively low inundation hazard may be attributable to the higher topographic elevations, lower rates of land subsidence and relatively low rates of geocentric SLR on the Pacific coast relative to the Atlantic and Gulf coasts[30]. Although the inundation hazard for Pacific coast communities (California's coast) by 2050 is relatively modest, rock coast cliff retreat[31] and the projected increase in the high-tide flooding[3] are further factors that would affect some coastal residents and properties.

## Land subsidence is a critical driver of coastal hazards

Land subsidence, the sinking or settling of the land surface, is a global issue with costly socioeconomic consequences[4,32,33]. To quantify the contribution of subsidence to future flooding hazards in the 32 US coastal cities, we consider two scenarios: (1) potential land areas below sea level resulting from land subsidence alone and (2) potential land areas below sea level resulting from a combination of land subsidence and sea-level change. Using linear projections of the current VLM rate and coastal-elevation data, we determine the land areas that, despite being above sea level at present, will be inundated by 2050 under both scenarios. Our analysis indicates that land areas below sea levels by 2050 resulting from only land subsidence account for 11.9–15.1% (Atlantic coast), 22.9–35.4% (Gulf coast) and 4.8–8.1% (Pacific coast) of total inundated areas when land subsidence and geocentric SLR are taken into consideration (Extended Data Fig. 5a–c and Supplementary Tables 5–7).

Furthermore, comparing our InSAR-derived VLM to geocentric sea-level change at Permanent Service for Mean Sea Level (PSMSL) tide-gauge stations included in the Intergovernmental Panel on Climate Change (IPCC) projections[17,34] across the USA (Supplementary Fig. 1) reveals that, at three stations on the Gulf coast, local land subsidence currently outpaces geocentric SLR (considering a low-emission scenario: SSP1-1.9) and presents a greater concern for the region than geocentric sea-level change (considering a high-emission scenario: SSP5-8.5) to 2045 (Extended Data Fig. 5d) and 2070 (Extended Data Fig. 5e,f), assuming linearly continuous subsidence. This observation is notable because it brings land subsidence to the fore of coastal-hazard discussions and highlights it as a crucial index in coastal disaster resilience design[20].

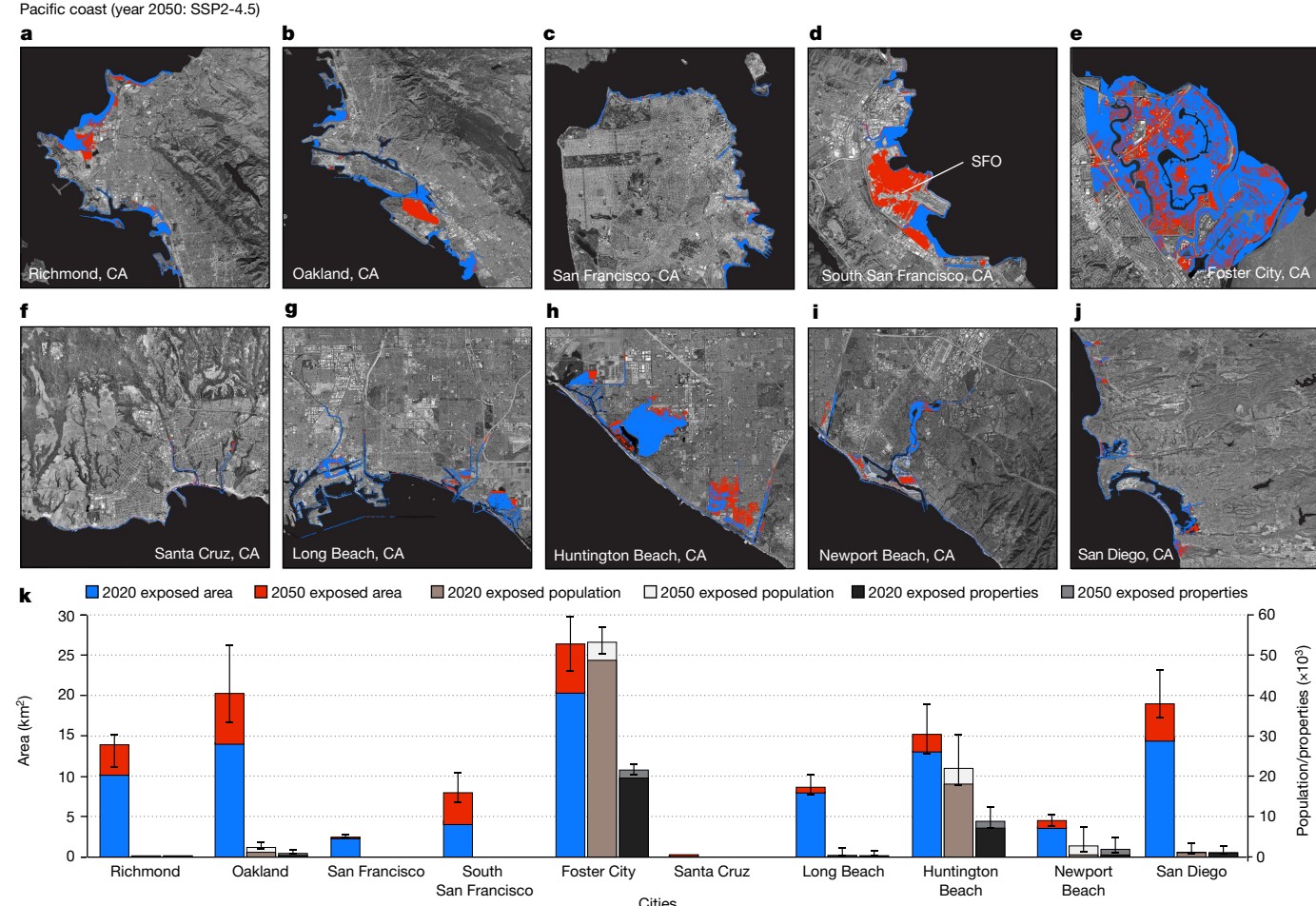

Pacific coast (year 2050: SSP2-4.5)

**Fig. 4 | Inundation maps for cities on the US Pacific coast.** Areas exposed to current (2020) high tide and further exposed areas by 2050 considering VLM, geocentric SLR projection under SSP2-4.5 for: Richmond, CA (**a**); Oakland, CA (**b**); San Francisco, CA (**c**); South San Francisco, CA (**d**); Foster City, CA (**e**); Santa Cruz, CA (**f**); Long Beach, CA (**g**); Huntington Beach, CA (**h**); Newport Beach, CA (**i**); and San Diego, CA (**j**). Background images in **a**–**j** are from Google, Earthstar.

Note that, for current exposure, geocentric SLR = 0 m. The blue and red colours are the current and projected exposed areas, respectively. SFO, San Francisco International Airport, CA. **k**, Distribution of further exposed areas, population and properties by 2050. The central value represents the estimated median value from equation (3), whereas the error bars represent the lower and upper bounds from equation (3).

Some flooding projections, however, do not consider the impacts of spatially variable land-elevation changes, resulting in inaccurate projections of expected exposure, which may affect the preparedness of coastal communities. For instance, a comparison of the estimated exposure by 2050 for the 32 cities derived for this study with exposure derived using the IPCC relative sea-level projections[34] shows that the exposure—in terms of area, population and properties—is not distinguishable within uncertainty in most of the cities and broadly for the entire US coast (Table 1). However, we find that notable divergence in the exposure occurs in some cities, particularly on the US Gulf coast (Fig. 5a–c and Supplementary Tables 8–10). By comparing the InSAR VLM rates within the cities with the IPCC-derived VLM rates from tide-gauge records, we gain insight into the underlying differences (Fig. 5d and Supplementary Table 11). In cities such as Boston, New York City, Naples, Port Arthur, Corpus Christi, Richmond, Oakland and San Francisco, in which IPCC VLM rates are similar to the InSAR VLM rates within the city or where the contribution of VLM is minimal, the disparities between the estimated exposure is modest. However, the estimated exposure is underestimated in cities in which the IPCC VLM underestimates the contribution of VLM (for example, Atlantic City, Charleston, Savannah, Mobile and Biloxi). Similarly, in cities in which the IPCC VLM overestimates the contribution of VLM, the estimated exposure is also overestimated (for example, Virginia Beach,

Jacksonville, Miami, New Orleans, Slidell, Lake Charles, Port Arthur and Huntington Beach).

To provide a more comprehensive understanding of these underlying differences, we performed a comparative analysis of InSAR-derived VLM rates with the VLM rates used in the IPCC Sixth Assessment Report[34] for 74 stations along the US coast (Supplementary Table 12 and Extended Data Fig. 6). Despite a 59% consistency in VLM measurements at the tide-gauge stations across the USA, these stations—often situated at the peripheries of urban areas—may not accurately capture the contemporary dynamics of spatially variable VLM within the cities themselves (for example, Biloxi, New Orleans and San Diego) (compare stations in Extended Data Fig. 6 with Fig. 5d). This limitation is particularly pertinent in urban centres in which anthropogenic drivers strongly influence VLM, thereby contributing to discrepancies in estimated exposure. InSAR observations are the gold standard of VLM measurements with unprecedented spatial resolution and are useful to enhance the accuracy of flood-prediction models.

## Climate-change inequalities

Climate change contributes to and exacerbates the fragility of the most vulnerable communities. Generally, disadvantaged populations are compelled to live in the most susceptible regions because safer areas

**Table 1 | Modelled further exposed area, population, properties and home-value exposure for the US coasts by 2050**

| Coast | | InSAR-derived | | | | IPCC-derived | | |
|---|---|---|---|---|---|---|---|---|
| | | 2050 further exposed area (km²) | 2050 further exposed population | 2050 further exposed properties | 2050 home-value exposure (US$ billion) | 2050 further exposed area (km²) | 2050 further exposed population | 2050 further exposed properties |
| Atlantic | a | 772.5 | 59,276 | 32,986 | 14.0 | 763.9 | 61,715 | 34,803 |
| | b | 871.5 | 100,276 | 60,580 | 25.0 | 871.3 | 96,866 | 58,658 |
| | c | 951.1 | 262,926 | 163,533 | 64.0 | 952.6 | 242,139 | 151,597 |
| Gulf | a | 536.7 | 110,647 | 58,423 | 14.0 | 663.3 | 203,896 | 99,421 |
| | b | 669.7 | 159,776 | 78,609 | 16.0 | 797.6 | 252,320 | 122,039 |
| | c | 827.6 | 225,167 | 109,505 | 21.0 | 924.6 | 286,080 | 142,089 |
| Pacific | a | 19.8 | 6,478 | 3,038 | 4.5 | 16.4 | 9,989 | 4,547 |
| | b | 29.0 | 12,180 | 5,707 | 9.3 | 28.2 | 13,433 | 6,301 |
| | c | 40.2 | 30,798 | 15,110 | 22.0 | 32.3 | 21,034 | 10,749 |
| Total | a | 1,329.0 | 176,401 | 94,447 | 32.5 | 1,443.6 | 275,600 | 138,771 |
| | b | 1,570.2 | 272,232 | 144,896 | 50.3 | 1,697.1 | 362,619 | 186,998 |
| | c | 1,818.9 | 518,891 | 288,148 | 107.0 | 1,909.5 | 549,253 | 304,435 |

a, b and c represent the lower bounds, median values and upper bounds evaluated using equation (3). The InSAR-derived exposure is estimated using InSAR VLM and IPCC geocentric SLR, whereas the IPCC-derived exposure is estimated using the IPCC relative SLR dataset[17,34]. See Supplementary Tables 2–4 for the InSAR-derived and Supplementary Tables 8–10 for the IPCC-derived exposure for each city.

are out of reach[35]. A growing body of scientific literature focuses on the impacts of climate change on vulnerable populations across countries[12,36,37]. However, less emphasis has been placed on discussions on within-country climate-change inequality[24,35]. A recent report by the United States Environmental Protection Agency (EPA)[38], along with the work of Hsiang et al.[24], contributes to the growing body of literature quantifying disproportionate climate-change risks to vulnerable communities in the USA.

Here we examine the disparate impacts of relative SLR on vulnerable communities across the 32 cities in the USA, offering a nuanced understanding of climate inequality in the USA. To examine the racial disparities in the exposed communities, we focused on eight races defined by the US census decennial data (see Methods). Our inundation model's analysis of the affected population by 2050 reveals differential exposure to relative SLR along racial lines (Extended Data Fig. 7 and Supplementary Tables 13–15). Along the Atlantic and Pacific coasts, white residents are overrepresented among those exposed to relative SLR (Extended Data Fig. 7a,c). However, the distribution of white exposed population on these coasts approximately reflects the dominant demographic of these regions (Extended Data Fig. 7a,c and Supplementary Tables 13 and 15).

By contrast, on the Gulf coast, minoritized groups—individuals identifying as Black or African American; American Indian or Alaska Native; Asian; Native Hawaiian or Other Pacific Islander; Hispanic or Latino; and two or more groups[38]—constitute a noteworthy portion of the exposed population, despite not being the dominant population. Although minoritized groups make up 43.0% of the total population across 11 cities in the Gulf coast, they are overrepresented in the exposed population, accounting for more than half (50.0–57.7%) of the exposed population in the case of Black or African American residents alone and 64.2–71.5% when considering all minorities (Extended Data Fig. 7b and Supplementary Table 14). Asians are overrepresented among the exposed population on the Gulf and Pacific coasts, accounting for 2.6–4.4% and 21.4–26.3% (median and upper bounds only) of the exposed population on the Gulf and Pacific coasts, respectively, despite making up 2.6% (Gulf coast) and 17.8% (Pacific coast) of the total population. Also, analysis of the impacts of relative SLR on economic inequality (see Methods) shows disproportionate economic exposure in New

Orleans and Port Arthur (Extended Data Fig. 8 and Supplementary Tables 16–18), which runs in parallel to the increased exposure of their predominantly minoritized communities. The intersection of racial and economic inequalities highlights the multidimensional vulnerability that specific populations in these cities face in the context of relative SLR.

One contributing factor to climate-change inequity is the diminished capacity of minoritized/low-income groups to adapt to and recover from the effects of existing hazards. To accentuate this point, consider the post-Hurricane Katrina impact on the population of Biloxi, MS. Although most attention is focused on the destruction to New Orleans caused by levee failures, Mississippi state incurred US$25 billion in direct storm damage and 100,000 people were displaced as a result of Hurricane Katrina[39]. East Biloxi, in which the low-income and marginalized community resides, was still suffering the impact of Hurricane Katrina 10 years after the storm, with broken, untended infrastructure, high unemployment rates and homelessness[37]. By contrast, the high-income communities received substantial federal aid for infrastructure rebuilding and are 'better off after the storm'[39]. This exclusion from recovery aid will undoubtedly impose constraints on the ability of low-income and minoritized communities to adapt to future climate change. Consequently, marginalized groups may account for a disproportionate share of climate-change collateral damage, even if they comprise a relatively small share of the affected population. In simple terms, existing inequalities make already vulnerable people more vulnerable to the adverse impacts of climate change, resulting in greater projected inequality[35].

## Towards sustainable adaptation strategies

Human interaction with the coast is increasing across the USA[10,40], and despite the clear and present danger posed by relative SLR, the need for adaptation and resilience planning is often overlooked or given insufficient priority. The growing risks identified in this paper pose a substantial challenge, and adaptation is to be expected in all the cities considered. Will cities be proactive and prepared through detailed assessment, planning and implementation of adaptive measures or will they be reactive to events, waiting for these impacts and risks to

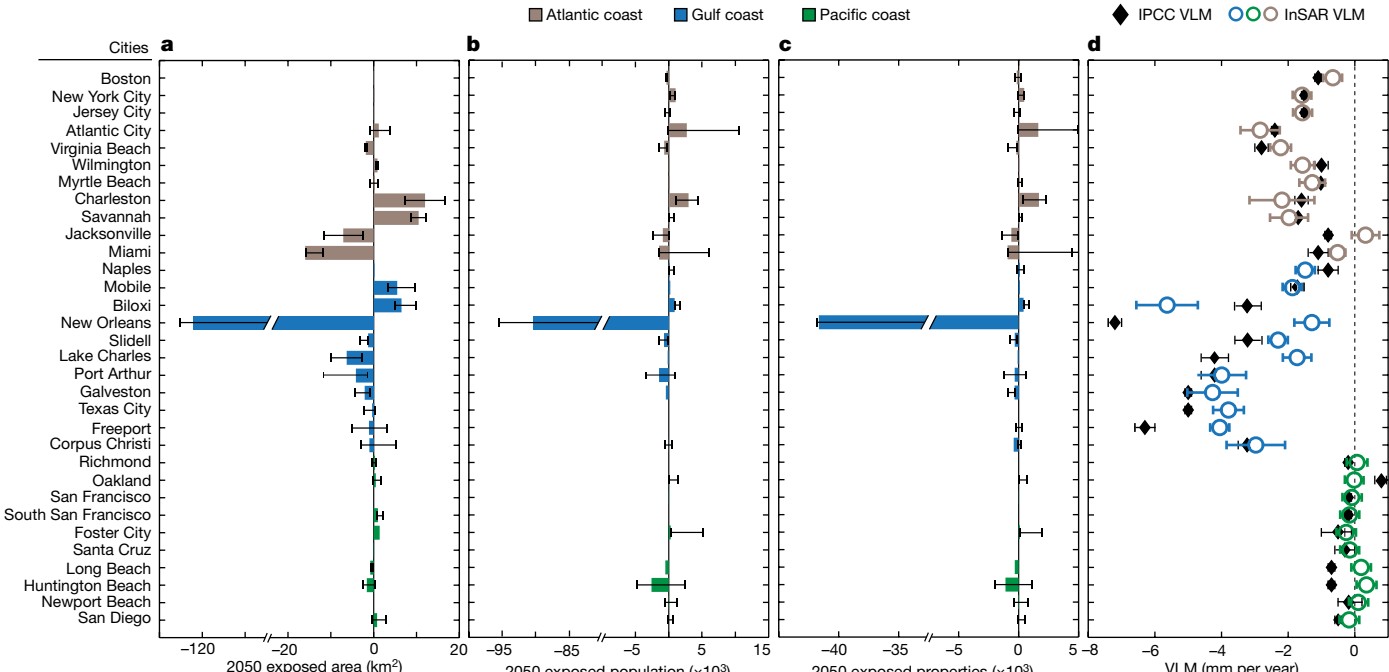

**Fig. 5 | Comparing InSAR-derived and IPCC-derived VLM and estimated exposure.** Differences between estimated exposure using InSAR-derived VLM and IPCC-derived VLM for 2050 exposed area (**a**), exposed population (**b**) and exposed properties (**c**). InSAR-derived exposure is estimated using InSAR VLM and IPCC geocentric SLR as detailed in Methods, whereas the IPCC-derived exposure is estimated using the IPCC relative SLR dataset[30]. The central value represents the estimated median value from equation (3), whereas the error bars represent the lower and upper bounds from equation (3). Negative values indicate cities in which IPCC-derived VLM exposure is overestimated, whereas positive values indicate areas in which exposure is underestimated. **d**, Comparison of InSAR versus IPCC VLM rates for the 32 cities. The InSAR VLM rates are obtained by averaging the VLM for each city used in the exposure analysis, whereas the IPCC VLM rates are derived from tide-gauge stations. The error ranges for the InSAR and IPCC VLM show ±1 standard deviation. A summary of the comparison of exposure and VLM rates is detailed in Supplementary Tables 8–11.

manifest? Here we will consider the proactive option. Adaptation is a long-term process and some combination of strategies is probably most appropriate sequenced following an adaptive-pathways approach[41]. The precise details will vary from place to place depending on the individual situation. An ideal case seems to be a combination of the following possibilities for coastal cities: maintenance of nature-based protection from marshes and mangroves; new and upgraded structural protection and land raising; subsidence control; and land-use planning to reduce vulnerability[42,43].

Artificial coastal-defence structures, such as levees, berms, dykes and floodwalls, protect coastal communities by decreasing the consequences of flooding and inundation in exposed areas[44]. Within the 32 coastal cities considered in this study, there are 131 flood-control structures, with more than 50% protecting the cities on the Pacific coast (Supplementary Table 19). To demonstrate the protective capacity of flood-control structures on the US coast to 2050, we modelled the exposed areas in all cities with at least one flood-control structure, considering a defended scenario (see Methods). Extended Data Fig. 9a–e depicts the spatial reduction in exposed areas and properties in some cities with levees and floodwalls. The defended scenario suggests that, by 2050, relative SLR will affect a land area of 1,006–1,389 km², 55,000–273,000 people and 31,000–171,000 properties on the US coasts, with 61–63% of the exposed area, 79–89% of the exposed population and 80–89% of the exposed properties being situated on the Atlantic coast (Extended Data Fig. 10 and Supplementary Tables 20–22). This ineffectiveness in hazard mitigation on the US Atlantic coast reflects the lack of an adequate flood-protection system in most cities. Ten cities on the US Atlantic coast evaluated in this study (excluding Miami) have only three levee systems (Extended Data Fig. 10d), opting for other protective approaches, such as beach nourishment, enhancing the beach and beachfront-property aesthetics, but providing more limited flood protection. Nevertheless, most existing structure-based coastal-defence systems were not designed with climate change in mind, and large upgrades may be required to remain effective even to 2050 (ref. 45). This is most relevant where subsidence and differential subsidence affect the use of flood-control structures by lowering their effective height below inundation depths and promoting structural failure[46] (Extended Data Fig. 9f–j).

Although not universal, human-induced land subsidence must also be mitigated where practical. Historically, land subsidence has been a silent problem with little public engagement or policy-focused studies, and its complex evolution and drivers make it a 'wicked' policy problem[47]. Although natural processes (for example, glacial isostatic adjustment (GIA)) influence coastal land subsidence on the US coasts, non-GIA processes, including anthropogenic subsidence caused by the accumulation of several shallow and deep subsurface activities, such as drainage, groundwater withdrawal and hydrocarbon extraction, at present contribute to relative SLR around the USA, particularly on the Atlantic and Gulf coasts[32,48] (Supplementary Figs. 2 and 3). Policies that aim to minimize subsidence (for example, through managed aquifer recharge) are crucial in the relevant cities[19,47,49].

Although the differences between low-emission and high-emission scenarios in terms of exposure are relatively modest in the short term, these variations are not inconsequential, given the inevitable continued rise in sea level beyond 2050 and its likely acceleration with further warming[9]. Therefore, a long-term proactive and continuous adaptation beyond simple coastal protection would be needed[50]. For the sustainability and resilience of US coastal cities, it is critical to adopt a built-upon multifaceted strategy involving the implementation of adaptive measures, the regulation of subsidence and the implementation of stringent climate-change policies that keep carbon emissions low. More importantly, these mitigation and adaptation strategies are

driven by anthropogenic influences and within reach with concerted societal efforts at all levels (policymakers to citizens).

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

## Methods

### VLM data

High-resolution VLM data are based on the Virginia Tech Earth Observation and Innovation (EOI) Lab VLM product, with spatially continuous coverage for the Pacific, Atlantic and Gulf coasts of the USA[27,28,51–53]. The dataset provides VLM measurements at millimetre-level accuracy and a resolution of about 50 m within a 100-km strip along the coasts of the USA. For each coast, the VLM rates were determined by integrating SAR images from Sentinel-1 A/B and ALOS-1 satellites between 2007 and 2020 (see Supplementary Table 23 for satellite frames used for each coast) with observations of horizontal and vertical velocities at global navigation satellite system (GNSS) stations. To produce the spatially continuous surface-deformation map, InSAR line-of-sight (LOS) displacements were generated for the numerous SAR frames along the coasts.

We use GAMMA software to process SAR datasets[54,55] and the wavelet-based InSAR (WabInSAR) algorithm to perform post-processing and multitemporal analysis[56–60] (Supplementary Fig. 4a). To this end, thousands of high-quality interferograms were generated and several wavelet-based analyses were applied to the interferograms to denoise the pixels and reduce the effects of spatially uncorrelated DEM error[56,57] and topographically correlated atmospheric phase delay[57]. Next, the velocity along the LOS direction for each pixel is calculated as the slope of the best-fitting line to the associated time series using a reweighted least-squares estimation. Last, the numerous SAR frames are mosaiced following Ojha et al.[61] and a stochastic model, which combines the LOS velocities with the GNSS datasets, was adopted to generate a high-resolution map of the VLM rate (Supplementary Fig. 4b).

To implement the stochastic model, we resampled the LOS velocities of Sentinel-1 tracks onto the ALOS track and interpolated the GNSS velocities on the pixels within the ALOS track using a Kriging interpolation technique with inverse distance weighting. Thus, we obtain several (5) observations per pixel for each coast, including LOS observations and GNSS velocities. Let $\{y_0, y_1,..., y_m\}$ and $\{\sigma_0^2, \sigma_1^2, ...\sigma_m^2\}$ be the interpolated LOS velocities and variances, respectively, for a given pixel, in which subscripts $0, 1,..., m$ indicate the available satellite observations (Sentinel-1/ALOS-1) and orbits (ascending/descending) for a given US coast (Atlantic/Gulf/Pacific), as defined in Supplementary Table 23. The stochastic model to combine the LOS velocities with the velocities of GNSS datasets to generate a seamless, high-resolution and accurate map of east (E), north (N) and vertical (U) motions is given by equation (1):

$$y_{0,1,...m} = C_e^{0,1,...m}E + C_n^{0,1,...m}N + C_u^{0,1,...m}U + \varepsilon^{0,1,...m}$$
$$E_G = E + \varepsilon^e$$
$$N_G = N + \varepsilon^n \qquad (1)$$
$$U_G = U + \varepsilon^u$$

in which $C$ represents the unit vectors projecting 3D displacements onto the LOS, which is a function of the satellite heading and incidence angles, $\varepsilon$ is the observation errors equal to the standard deviations ($\sigma$), $E$, $N$ and $U$ are the unknowns and $E_G$, $N_G$ and $U_G$ are the observed interpolated east, north and up GNSS velocities, respectively. The solution to equation (1) is given by equation (2):

$$X = (A^T PA)^{-1} A^T PL \qquad (2)$$

in which $X$ represents the unknowns, $A$ is the Green's function given by the unit vectors ($C$), $L$ is the observation and $P$ is the weight matrix, which is inversely proportional to the observant variance ($\sigma^2$). The parameters variance-covariance matrix is $Q_{XX} = \frac{r^T Pr}{df}(A^T PA)^{-1}$, in which $r = L - AX$ and df is the degrees of freedom. The standard deviations (precision of the results) for each pixel on the Atlantic/Gulf/Pacific

coasts are shown in Supplementary Fig. 5a. The spatial distribution of the standard deviation shows that most values are below 3 mm per year for the US Atlantic and Gulf coasts. However, there are a few hotspots of high standard deviation around the Chesapeake Bay area (US Atlantic coast) and around the coast of Florida (US Gulf coast). We note higher estimated standard deviation values in the US Pacific coast, specifically in Northern California and the Orange County basin[27] (Supplementary Fig. 5a). Generally, higher standard deviation values represent areas of lower precision. The observed lower precision in some pixels may be attributed to lower interferometric phase signal-to-noise ratio caused by surface vegetation, nonlinearity in the rates between the ALOS and Sentinel-1 observation periods owing to aquifer recharge and depletion, a limited number of GNSS stations used for the adjustment and a comparatively higher standard deviation of the GNSS station in the particular regions[20,27,28]. Furthermore, we validate the VLM rates using 756 GNSS stations (US Atlantic coast: 218; US Gulf coast: 157; US Pacific coast: 381) from the Nevada Geodetic Laboratory[62] and Shirzaei et al.[48]. To perform the validation, we computed the average InSAR VLM rates within a 200-m radius around each GNSS station for comparison with the corresponding GNSS vertical rates (Extended Data Figs. 2–4). We obtained a standard deviation of 1.5 mm per year and a mean difference of less than 0.3 mm per year for the US Atlantic, Gulf and Pacific coasts (Supplementary Fig. 5b–d).

Spatial analysis of the complied VLM map (Fig. 1 and Extended Data Figs. 2–4) reveals extensive coastal areas with subsidence rates of more than 3 mm per year. Figure 1e highlights the spatially variable VLM for the 32 major coastal cities selected for this study: US Atlantic coast: Boston, MA; New York City, NY; Jersey City, NJ; Atlantic City, NJ; Virginia Beach, VA; Wilmington, NC; Myrtle Beach, SC; Charleston, SC; Savannah, GA; Jacksonville, FL; Miami, FL; US Gulf coast: Naples, FL; Mobile, AL; Biloxi, MS; New Orleans, LA; Slidell, LA; Lake Charles, LA; Port Arthur, TX; Galveston, TX; Texas City, TX; Freeport, TX; Corpus Christi, TX; US Pacific coast: Richmond, CA; Oakland, CA; San Francisco, CA; South San Francisco, CA; Foster City, CA; Santa Cruz, CA; Long Beach, CA; Huntington Beach, CA; Newport Beach, CA; San Diego, CA.

We find subsidence rates greater than 2 mm per year in 24 out of 32 major cities along the US Atlantic, Gulf and Pacific coasts, with notable subsidence rates (>5 mm per year) in cities such as Charleston (city number 8), Biloxi (city number 14), Galveston (city number 20) and Corpus Christi (city number 22) (Fig. 1e). On the US Pacific coast, we observe lower rates of land subsidence compared with the Atlantic and Gulf coasts, with some cities characterized by marked uplift (such as Richmond: city number 23; Long Beach: city number 29; Huntington Beach: city number 30; and Newport Beach: city number 31).

Subsidence along the coast is driven by natural and human processes and is a notable contributor to relative sea-level change[2,19,48,49,63–65]. Earlier studies suggested that complex processes drive observed subsidence along the US coasts[27,66–69]. These drivers include a combination of natural and anthropogenic processes, such as GIA, compaction of sediments, groundwater withdrawal, hydrocarbon extraction, surficial drainage/dewatering activities and regional tectonic activities. On a broad scale, disentangling the contribution of natural-driven and anthropogenic-driven processes is important for developing effective strategies to mitigate or adapt to the impacts of subsidence in low-lying coastal cities. On the one hand, in cities in which subsidence is a result of GIA and other natural processes, effective subsidence mitigation will probably involve an adaptive response, such as raised structures and infrastructure and flood-protection measures. On the other hand, for anthropogenic processes, proactive policy interventions and mitigation measures to reduce and control resulting subsidence, such as reducing groundwater and oil and gas extraction or changes in land use, may be helpful in sinking cities. As GIA is the main natural driver, we used the GIA ICE-6G-D model[70] to estimate the GIA contributions at the SAR pixels and subtracted its effect from the observed VLM to

assess the non-GIA contributions to the estimated VLM along US coasts (Supplementary Fig. 2). The relative reduction of subsidence by 46%, 4% and 20% for the Atlantic, Gulf and Pacific coasts, respectively, suggests that the effect of GIA on subsidence is dominant primarily along the US Atlantic coast and minimal for the Gulf and Pacific coasts (Supplementary Fig. 2c–e). Although the median rates of subsidence are reduced for all 32 major coastal cities, several areas with subsidence rates greater than 2 mm per year remain apparent in more than half of the selected cities, such as Boston, Atlantic City, Charleston, Biloxi, New Orleans, Texas City, San Francisco, Foster City and San Diego (Supplementary Fig. 3).

### Coastal cities selection and elevation data

To select the 32 cities for analysis, we considered 41 major US coastal cities with VLM and LiDAR DEMs data. We conducted a preliminary analysis to determine the exposed area of each city, considering the IPCC localized (relative) SLR projections and mean high water (MHW) of the nearest tide gauge. Next, we screened the cities on the basis of the largest exposed area and selected ten cities from each US coastal region as follows: US Atlantic coast: Boston, MA; New York City, NY; Atlantic City, NJ; Virginia Beach, VA; Wilmington, NC; Myrtle Beach, SC; Charleston, SC; Savannah, GA; Jacksonville, FL; Miami, FL; US Gulf coast: Naples, FL; Mobile, AL; Biloxi, MS; New Orleans, LA; Slidell, LA; Lake Charles, LA; Port Arthur, TX; Texas City, TX; Freeport, TX; Corpus Christi, TX; US Pacific coast: Richmond, CA; Oakland, CA; San Francisco, CA; South San Francisco, CA; Foster City, CA; Santa Cruz, CA; Long Beach, CA; Huntington Beach, CA; Newport Beach, CA; San Diego, CA.

On the Pacific coast, we focused only on future inundation hazards for cities in California. The absence of coastal cities from Oregon and Washington is a result of the lack of high-resolution VLM data for the US northwest coast (Fig. 1) and the complexities of future inundation hazards in the region driven by earthquake and tsunami hazards. The aftermath of earthquake and tsunami hazards can cause substantial subsidence followed by inundation from tsunami waves. Evaluating such hazards requires a probabilistic analysis of future earthquake and tsunami hazards beyond this study.

Also, we added two cities, Jersey City, NJ and Texas City, TX that were located near other selected cities (New York City, NY and Galveston, TX, respectively) and are also important urban centres in their respective regions.

We use LiDAR DEM for the coastal-elevation data. The high-resolution LiDAR DEMs hosted by the National Oceanic and Atmospheric Administration (NOAA) Office for Coastal Management are available for the coastal USA[71]. In this study, we used a 3-m × 3-m grid resolution for the 32 cities, except Savannah (GA), Jacksonville (FL), Miami (FL) and all cities on the Pacific coast, which were obtained at a 5-m × 5-m grid resolution (Supplementary Table 23). All DEMs for each city use the North American Vertical Datum of 1988 (NAVD 88) vertical datum. Details on the implementation, vertical and horizontal accuracy, errors and temporal range are available with the data download[71].

### Population, properties and racial demographic data

We estimate the population and property datasets for each city using the Topologically Integrated Geographic Encoding and Referencing (TIGER) system demographic and economic data records available from the US Census Bureau (https://www.census.gov/geographies/mapping-files/time-series/geo/tiger-data.2010.html). The dataset provides population and property estimates for each city in the USA, subdivided into census blocks based on the 2010 census data. We used the 2010 dataset because it is the most recent census data available from the US Census Bureau. The racial demographic dataset is based on the decennial Census data (https://data.census.gov/cedsci/advanced) corresponding to the 2010 census. For this study, we selected eight races: 'White', 'Black or African American', 'American Indian and Alaska Native', 'Asian', 'Native Hawaiian and Other Pacific Islander', 'Hispanic or Latino' and others ('Some Other Race' alone and 'Two or More Races'), as defined by the decennial data.

### Sea-level projections

We use the localized sea-level projections from the IPCC Sixth Assessment Report[17,34]. The projections consider the contributions to future sea levels from sterodynamic effects (ocean steric and ocean dynamic effects), ice sheets (Antarctic and Greenland ice sheets), land water storage, glacier and ice cap surface mass balance, thermal expansion and IPCC estimates of total VLM based on tide-gauge observations—reflecting the sum of GIA and other VLM processes. To prevent double counting of VLM, we acquired the SLR projections without the effect of VLM for our analysis (geocentric SLR). The database provides projections of sea level at tide-gauge stations worldwide under five SSP scenarios (SSP1-1.9, SSP1-2.6, SSP2-4.5, SSP3-7.0 and SSP5-8.5). SSP1-1.9 limits warming to 1.5 °C above 1850–1900 levels by 2100, implying net-zero $CO_2$ emissions around the middle of the century. SSP1-2.6 keeps warming below 2.0 °C relative to 1850–1900, with projected net-zero emissions in the second half of the century. The SSP2-4.5 scenario projects best-estimate warming of approximately 2.7 °C by the end of the twenty-first century relative to 1850–1900. SSP3-7.0 is a medium to high reference scenario resulting from no further climate policy with particularly high non-$CO_2$ and aerosol emissions and a warming of 2.8–4.6 °C. SSP5-8.5 is a high reference scenario with the highest emission levels (above the current emissions trajectory) and warming of 3.3–5.7 °C. In this study, we apply the 17th (lower bound), 50th (median) and 83rd (upper bound) percentile projections under SSP2-4.5, which represents the current emissions trajectory. Supplementary Table 24 summarizes the tide-gauge stations used for the SLR projections in each city.

### High-tide estimates

We used MHW at tide gauges to estimate the high-tide events for each coastal city. The tide-gauge measurements were obtained from NOAA tide and currents data[72], using the NAVD 88 datum, consistent with the elevation datum and mean measurement for the present epoch. Tide-gauge stations used for each city were selected on the basis of the proximity to the city, which provides localized data crucial for accurate evaluation of the current exposure to high tide in the urban areas (Supplementary Table 24).

### Inundation model

Using a bathtub model[2,51,73–75] (see Supplementary Fig. 4c), we projected the inundation hazards for 32 cities on the US coasts. The input data for the inundation model are as follows:
1. 3-m or 5-m grid LiDAR DEM for each city.
2. About 50-m resolution VLM data for each city.
3. MHW levels at tide gauges adjacent to each city (Supplementary Table 24).
4. IPCC geocentric SLR projections at the stations adjacent to each city (Supplementary Table 24).

To provide a comprehensive exposure assessment, we incorporate two temporal scales—the current (2020) and projected exposure (2050). First, the current exposure to high tide is assessed using MHW levels. Subsequently, projected exposure is evaluated by considering both VLM and geocentric SLR. Thus, the projected exposure represents further exposure, providing a baseline of current exposure against which future scenarios can be compared. To implement the inundation model, first, we resample the VLM rates on the LiDAR DEM. Next, we modify the elevation model to account for VLM projections, assuming a linear VLM rate from the base year of the DEM to the target years of 2020 and 2050 (refs. 34,48,51). Last, we evaluated the current (2020) scenario by subtracting the modified DEM height, which accounts for

VLM projections up to the year 2020, from MHW levels (equation (3)). Subsequently, for the 2050 scenario, we apply SLR projections by subtracting the modified DEM height, updated for VLM projections to the year 2050, from both the geocentric SLR projection height and the MHW levels (equation (3)). Areas with a projected height below zero are inundated. This simplified static model is useful for local-scale simulations of inundated locations hydrologically connected to the coast[74]. However, it may overestimate or underestimate inundated areas on the coast owing to the reduced complexity of the model[74]. To reduce the errors associated with this approach, we implemented connected-component analysis to remove solitary grid cells from the inundation model, which represents topographically isolated low regions. Furthermore, we first present our inundation model as an undefended inundation map that does not account for the presence of levees or sea walls and we introduce and discuss the possible implications of flood-control structures on the impacts of relative SLR. For the defended scenario, we account for existing levees and sea walls by modifying the DEM height at the location of flood-control structures above the threshold for potential flooding.

To account for all error sources in the input data, we consider the uncertainties in the DEM, VLM and SLR datasets. Specifically, we propagate the 17th and 83rd percentiles for geocentric SLR projections, ±1 standard deviation for VLM and errors inherent in the DEM (equation (3)). These measures provide an error bound for the inundation analysis, ensuring robust estimation of the uncertainties associated with the projections.

$$Inun_{med} = DEM_{mod} - (SLR_{50} + MHW)$$

$$Inun_{low,up} = Inun_{med}$$

$$\pm \left( \sqrt{DEM_{err}^2 + ((t-t_0)VLM_{SD})^2 + ((SLR_{83} - SLR_{17})/2)^2} \right) \quad (3)$$

in which $Inun_{low}$, $Inun_{med}$ and $Inun_{up}$ represent the median, lower and upper bounds, respectively, for the inundation models. $DEM_{mod}$ is the modified DEM height, updated using the VLM projections. $DEM_{err}$ is the vertical accuracy of the DEM. $t$ represents the projection target years of 2020 or 2050. $t_0$ represents the base year of the DEM. $VLM_{SD}$ is one standard deviation from the VLM data. MHW represents mean high water. $SLR_{17}$, $SLR_{50}$ and $SLR_{83}$ represent the 17th, 50th and 83rd percentiles, respectively, from the geocentric SLR projections. Note that, for evaluating the current/baseline scenario (2020), $SLR_{17}$, $SLR_{50}$ and $SLR_{83}$ are zero.

## Socioeconomic exposure analysis

We used the TIGER demographic and economic data to assess the population and property exposure, which estimates the total population and properties subdivided into census blocks. We consider a census block inundated if greater than 20% of its area is inundated and assign the population and property for that block as exposed population or properties. To select the 20% threshold for the exposure of each census block, we conducted an empirical analysis across six representative cities. The distribution of exposed areas within these census blocks followed an extreme-value distribution. Statistical metrics revealed a median value of exposed area ranging from 18% to 23% for the six cities (Supplementary Fig. 6). Furthermore, the distribution exhibited a sharp decline beyond 10% (Supplementary Fig. 6). Therefore, a 20% criterion was established as a suitable threshold for quantifying the exposed population and properties. To quantify the home-value exposure, we used the ZIP Code ZHVI as a metric for housing cost. The estimated home-value exposure was calculated by multiplying the number of exposed properties within each city by the corresponding ZHVI (https://www.zillow.com/research/data/). We adjusted the ZHVI for the recent economic inflation using the mid-2021 housing price. The Zillow home-value data for each ZIP code used in this study is reported in Supplementary Table 25.

To investigate the disparate sociodemographic and socioeconomic impacts of relative SLR on vulnerable groups, we focused on analysing both racial and economic disparities in the exposed communities. To examine the racial disparities in the exposed communities, we considered eight races as defined by the US census decennial data: 'White', 'Black or African American', 'American Indian and Alaska Native', 'Asian', 'Native Hawaiian and Other Pacific Islander', 'Hispanic or Latino', 'Some Other Race' alone and 'Two or More Races'. Racial minoritized groups are defined as individuals identifying as Black or African American; American Indian or Alaska Native; Asian; Native Hawaiian or Other Pacific Islander; Hispanic or Latino; and two or more groups[38]. The analysis shows an overrepresentation of the white population on the Atlantic and Pacific coasts, whereas minoritized populations are overrepresented on the Gulf coast. On the Atlantic coast, the white population makes up 55.1–71.4% of the exposed population by 2050, which is higher than their 38.3% share in the total population (Extended Data Fig. 7 and Supplementary Table 13). Minoritized groups make up 43.0% of the total population on the Gulf coast while accounting for 64.2–71.5% of the exposed population by 2050 (Extended Data Fig. 7 and Supplementary Table 14). On the Pacific coast, white residents comprise 57.6–70.9% of the exposed population by 2050, despite making up only 41.8% of the total population (Extended Data Fig. 7 and Supplementary Table 15). In cities such as Jersey City, New Orleans, Port Arthur and Oakland, minority groups are disproportionately represented among the exposed population (Supplementary Tables 13–15).

To assess the impacts of relative SLR on economic inequality, we used the property value as a proxy for economic status. Using a Kolmogorov–Smirnov statistical method, we compare the median home values in regions exposed to relative SLR by 2050 against those in each city, using an alpha value of 0.05 for statistical significance. Supplementary Tables 16–18 summarize the statistical test for the 32 cities. Note that 9–22 cities (considering lower to upper bound relative SLR projections) were excluded from the analysis because of limitations imposed by the central limit theorem. Across the 14 cities examined (considering median relative SLR projection), we find statistically significant economic disparities in 12 cities (Extended Data Fig. 8). In eight of these cities, we find that the median home value for the exposed population is higher than the total home value in the cities (that is, exposed properties are overvalued). However, in Atlantic City, New Orleans, Port Arthur and Foster City, we find that the median exposed-home values are lower than the overall median home value within the cities, highlighting their disproportionate economic exposure.

## Subsidence hazard exposure analysis for levees

The polygon features for the levees across the US coasts were obtained from the United States Army Corps of Engineers (USACE)[76]. To determine the exposure to subsidence for the levees, we extracted the VLM rate for each point along the polygon feature. The subsidence exposure for levees in five cities (Miami, FL; New Orleans, LA; Port Arthur, TX; Freeport, TX; and Foster City, CA) are shown in Extended Data Fig. 9.

## Data availability

The VLM data for the Pacific coast are available through the Virginia Tech Data Repository at https://doi.org/10.7294/17711000. The VLM data for the Atlantic coast are available through the Virginia Tech Data Repository at https://doi.org/10.7294/19350959. The VLM data for the Gulf coast are available through the Virginia Tech Repository at https://doi.org/10.7294/22731326. The supplementary tables for this manuscript are made accessible through the Virginia Tech Data Repository at https://doi.org/10.7294/24782199. The levee dataset is available from the United States Army Corps of Engineers (USACE) National Levee Database (https://levees.sec.usace.army.mil). The population and properties datasets are available from the US Census Bureau (https://www.census.gov/). The racial demographic dataset is based on the Decennial

Census data (https://data.census.gov/cedsci/advanced). The housing value is obtained from the Zillow Home Value Index (ZHVI) (https://www.zillow.com/research/data/). The coastal elevation data are LiDAR digital elevation data hosted by the National Oceanic and Atmospheric Administration (NOAA) (Digital Coast: Data Access Viewer: https://coast.noaa.gov/dataviewer/#/lidar/search/). The high-tide data are obtained from Tides and Currents: Datums from NOAA (https://tidesandcurrents.noaa.gov/stations.html?type=Datums). All other data needed to evaluate the conclusions are presented in the paper and the supplementary materials.

## Code availability

The WabInSAR code used to perform the synthetic aperture radar (SAR) analysis is available at https://sites.google.com/vt.edu/eadar-lab/software?authuser=0.

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

**Acknowledgements** We thank S. A. Talke for discussing the high-tide data. L.O.O. is supported by a U.S. Geological Survey grant. M.S. is supported by U.S. Geological Survey and National Science Foundation grants.

**Author contributions** L.O.O. and M.S. designed the research and created the figures. L.O.O., M.S. and C.O. performed the analysis. L.O.O., M.S. and R.J.N. wrote the first draft of the paper. L.O.O., M.S., C.O., S.F.S. and R.J.N. analysed the results and edited the paper.

**Competing interests** The authors declare no competing interests.

**Additional information**
**Correspondence and requests for materials** should be addressed to Leonard O. Ohenhen.

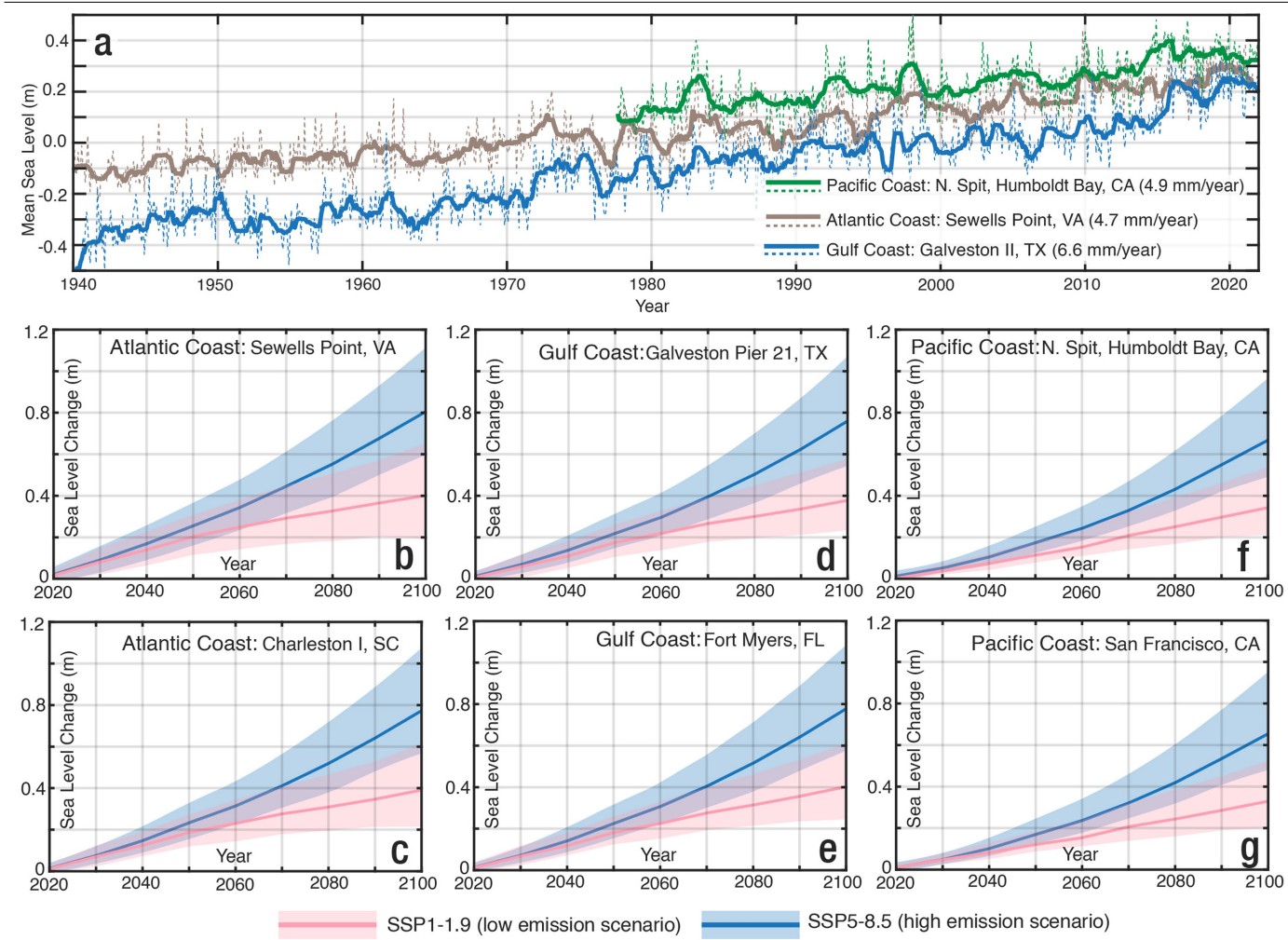

**Extended Data Fig. 1 | Current and projected SLR at selected tide-gauge stations across the US coasts. a**, Local mean sea level measured at tide gauges for monthly (dashed line) and annual (solid line) time series. For clarity, the time series for the Pacific and Atlantic coasts have been offset by factors of +0.2 and +0.1, respectively. The corresponding tide-gauge locations for the projected sea-level change and time-series data are the green, brown and blue inverted triangles shown in Fig. 1a. Comparison of low-emission scenarios (SSP1-1.9) with high-emission scenarios (SSP5-8.5) (medium confidence) for the Atlantic coast Sewell's Point, VA (**b**) and Charleston I, SC (**c**); Gulf coast Galveston Pier 21, TX (**d**) and Fort Myers, FL (**e**); and Pacific coast North Spit, Humboldt Bay, CA (**f**) and San Francisco, CA (**g**).

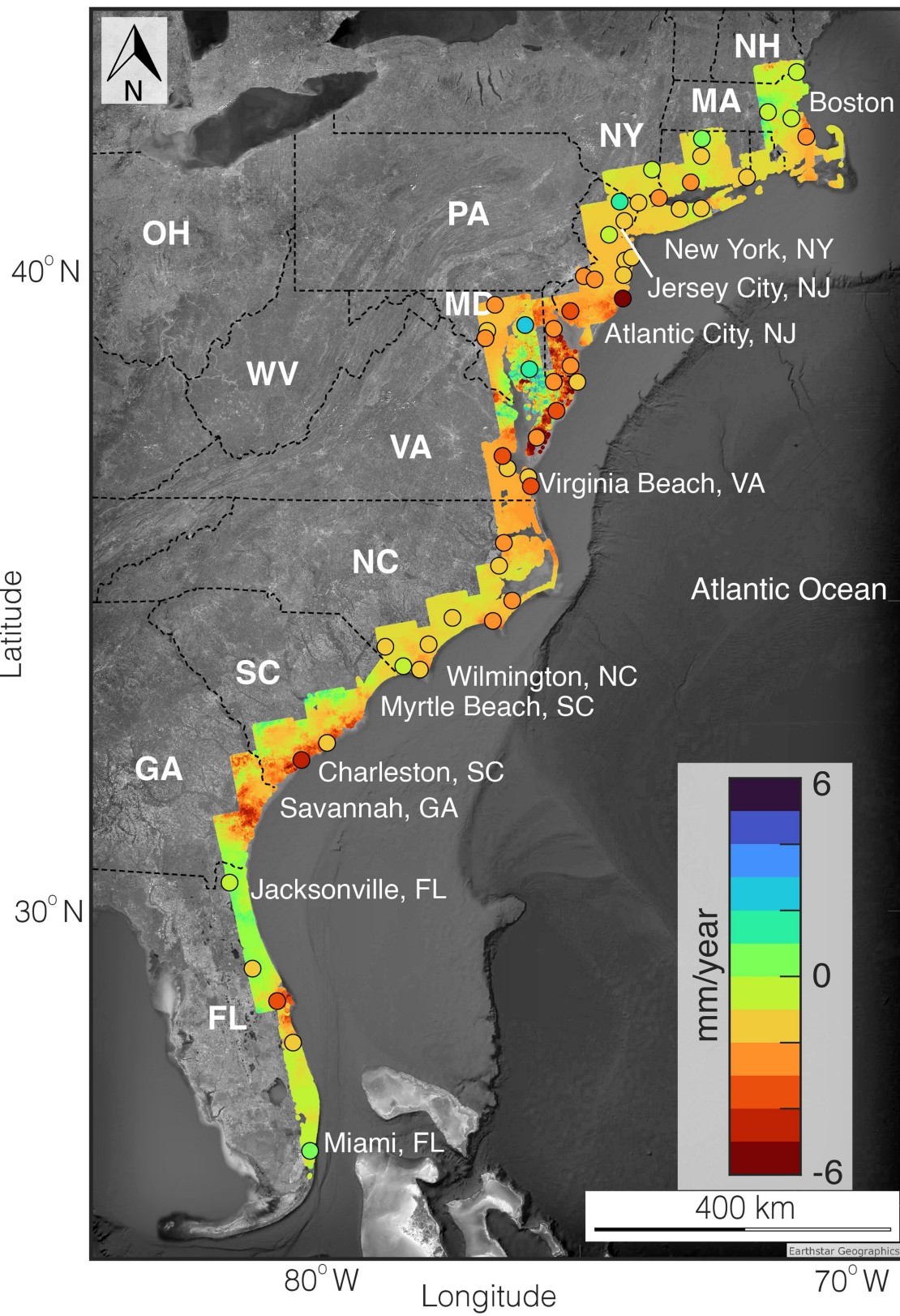

**Extended Data Fig. 2 | Spatial distribution of VLM across the US Atlantic coast.** Positive VLM rates indicate an uplift and negative VLM rates indicate subsidence. The 11 US Atlantic coastal cities evaluated in this study are highlighted in the figure. Background image is from Google, Earthstar. The vertical velocities of GNSS validation stations are shown using colour-coded circles on the map. Note that only a subset of the GNSS data is plotted to prevent cluttering. The comparison of vertical rates from 218 GNSS stations with InSAR rates is shown in Supplementary Fig. 11c. State codes: NH, New Hampshire; MA, Massachusetts; NY, New York; PA, Pennsylvania; NJ, New Jersey; MD, Maryland; WV, West Virginia; OH, Ohio; VA, Virginia; NC, North Carolina; SC, South Carolina; GA, Georgia; FL, Florida. State boundaries are based on public-domain vector data by the World Bank DataBank (https://data.worldbank.org/) generated in MATLAB.

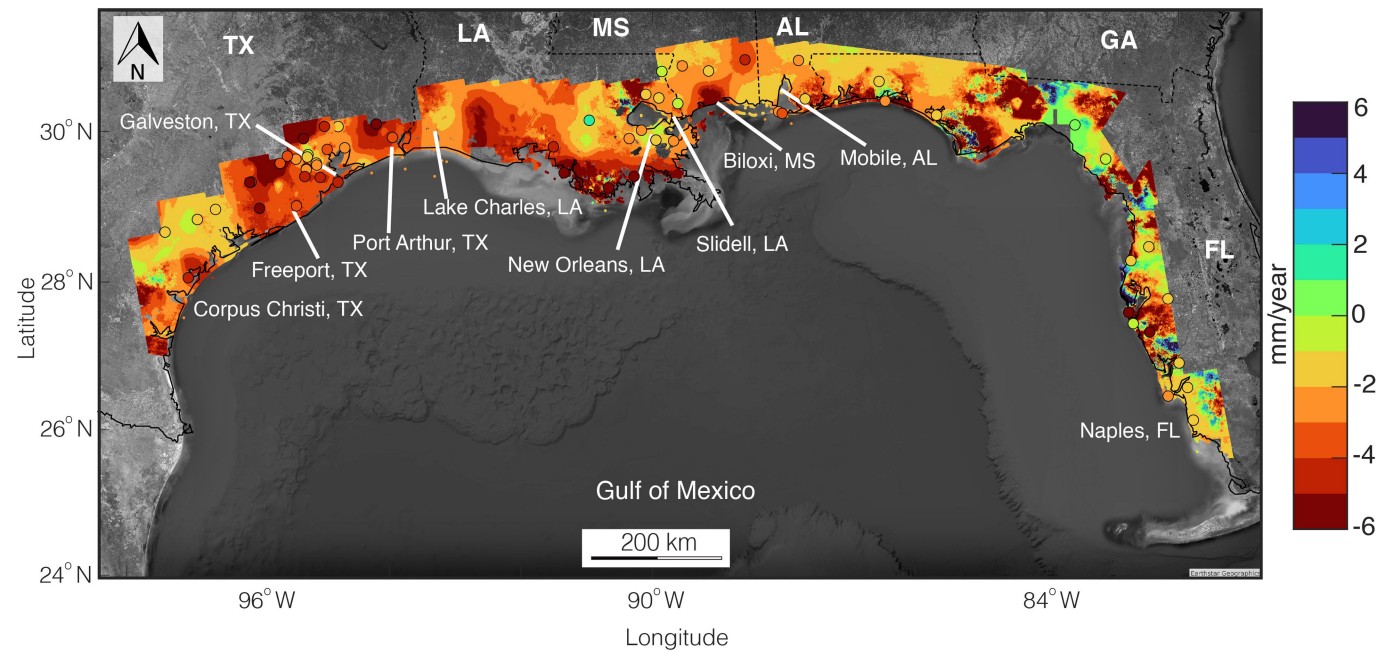

**Extended Data Fig. 3 | Spatial distribution of VLM across the US Gulf coast.** Positive VLM rates indicate an uplift and negative VLM rates indicate subsidence. The 11 US Gulf coastal cities evaluated in this study are highlighted in the figure. Background image is from Google, Earthstar. The vertical velocities of GNSS validation stations are shown using colour-coded circles on the map. Note that only a subset of the GNSS data is plotted to prevent cluttering. The comparison of vertical rates from 157 GNSS stations with InSAR rates is shown in Supplementary Fig. 11d. State codes: TX, Texas; LA, Louisiana; MS, Mississippi; AL, Alabama; GA, Georgia; FL, Florida. National and state boundaries are based on public-domain vector data by the World Bank DataBank (https://data.worldbank.org/) generated in MATLAB.

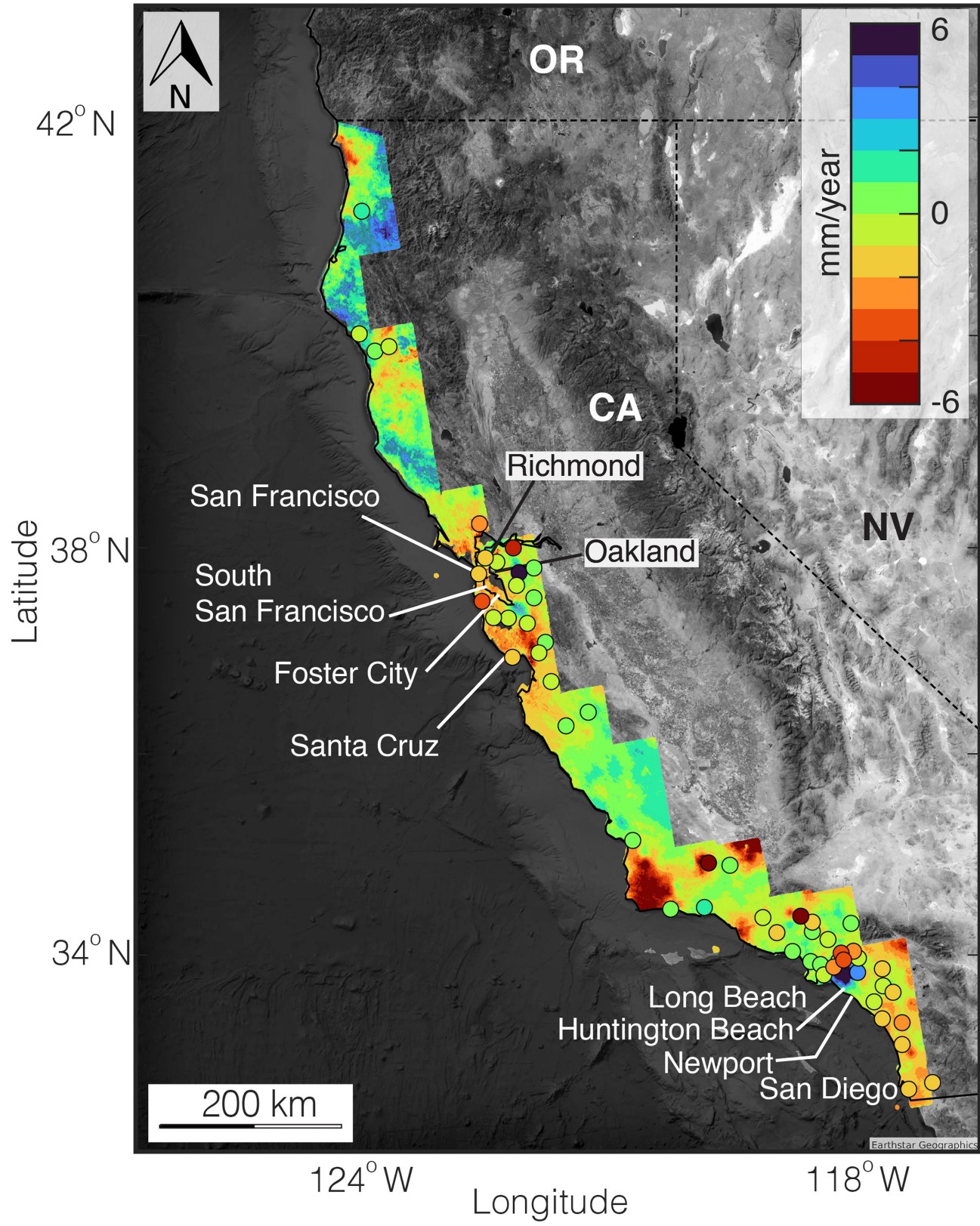

**Extended Data Fig. 4 | Spatial distribution of VLM across the US Pacific coast.** Positive VLM rates indicate an uplift and negative VLM rates indicate subsidence. The ten US Pacific coastal cities evaluated in this study are highlighted in the figure. Background image is from Google, Earthstar. The vertical velocities of GNSS validation stations are shown using colour-coded circles on the map. Note that only a subset of the GNSS data is plotted to prevent cluttering. The comparison of vertical rates from 381 GNSS stations with InSAR rates is shown in Supplementary Fig. 11b. State codes: OR, Oregon; CA, California; NV, Nevada. National and state boundaries are based on public-domain vector data by the World Bank DataBank (https://data.worldbank.org/) generated in MATLAB.

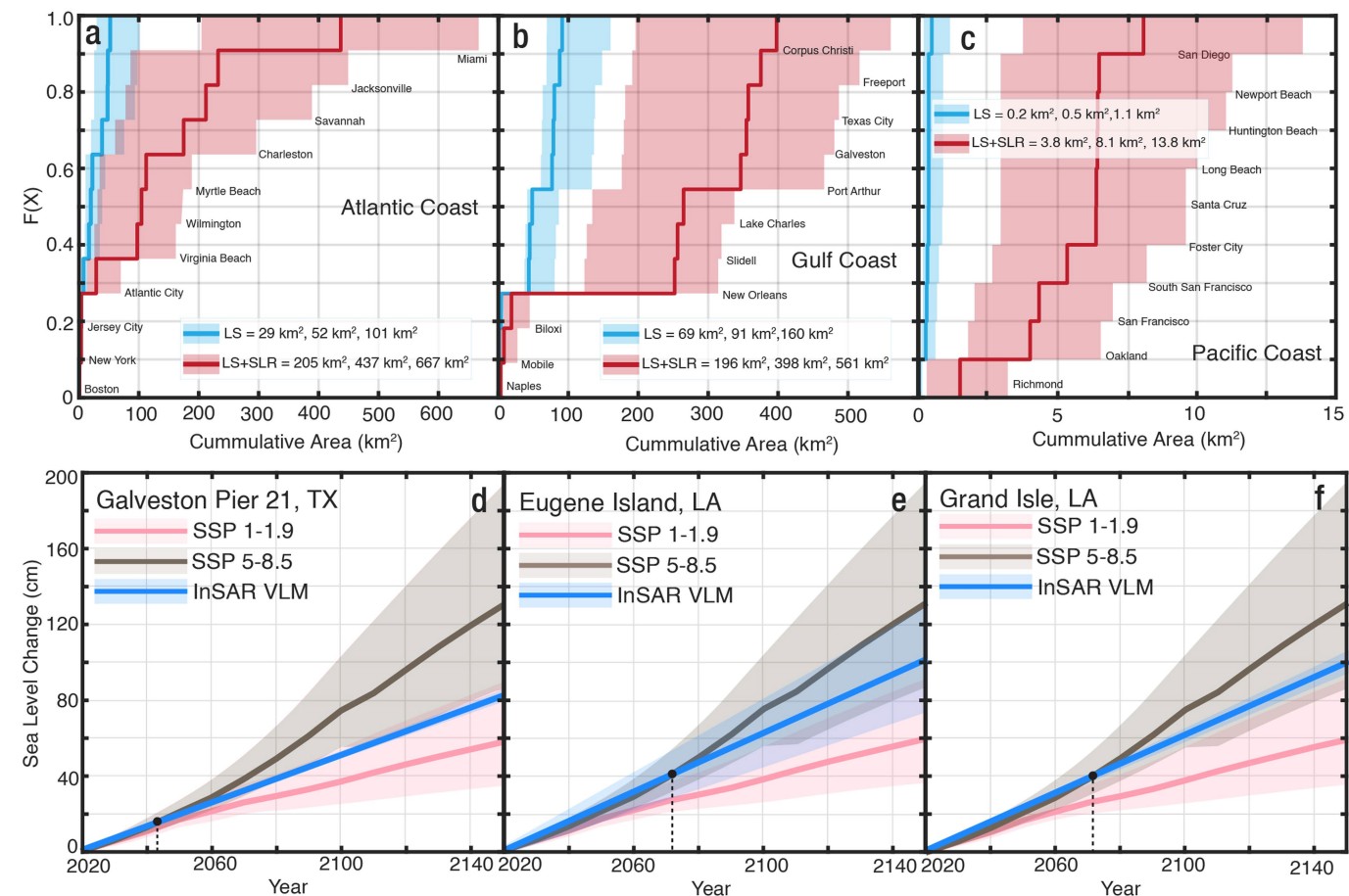

**Extended Data Fig. 5 | Contribution of land subsidence.** Empirical cumulative distribution function of the total exposed area by 2050 for the US Atlantic coast (**a**), Gulf coast (**b**) and Pacific coast (**c**). The blue lines and shaded area show the contribution from land subsidence (LS) alone, considering linear projection of VLM from 2020 to 2050. The red lines and shaded area show the contribution from land subsidence and 2050 SLR under SSP2-4.5. The red lines in **a**–**c** represent the median value considering equation (3), whereas the shaded region is the lower and upper bounds from equation (3). The total exposed area for each coast are the values in **a**–**c**. Comparison of geocentric sea-level change from the IPCC Sixth Assessment Report[17,34], with sea-level change resulting from land subsidence from InSAR measurements in this study

at three tide-gauge stations on the Gulf coast: Galveston Pier 21, TX (**d**); Eugene Island, LA (**e**); and Grand Isle, LA (**f**). The solid red line shows the median (50th percentile) IPCC sea-level change under SSP1-1.9 (low-emission scenario), whereas the red shaded range shows the 17th–83rd percentile. The solid brown line shows the median (50th percentile) IPCC sea-level change under SSP5-8.5 (high-emission scenario), whereas the brown shaded range shows the 17th–83rd percentile. The solid blue line shows the InSAR VLM from this study and the blue shaded ranges are one standard deviation. The black point and dashed line indicate when other processes exceed sea-level change from land subsidence. An example comparison for 20 tide-gauge stations across the US subsidence is shown in Supplementary Fig. 1.

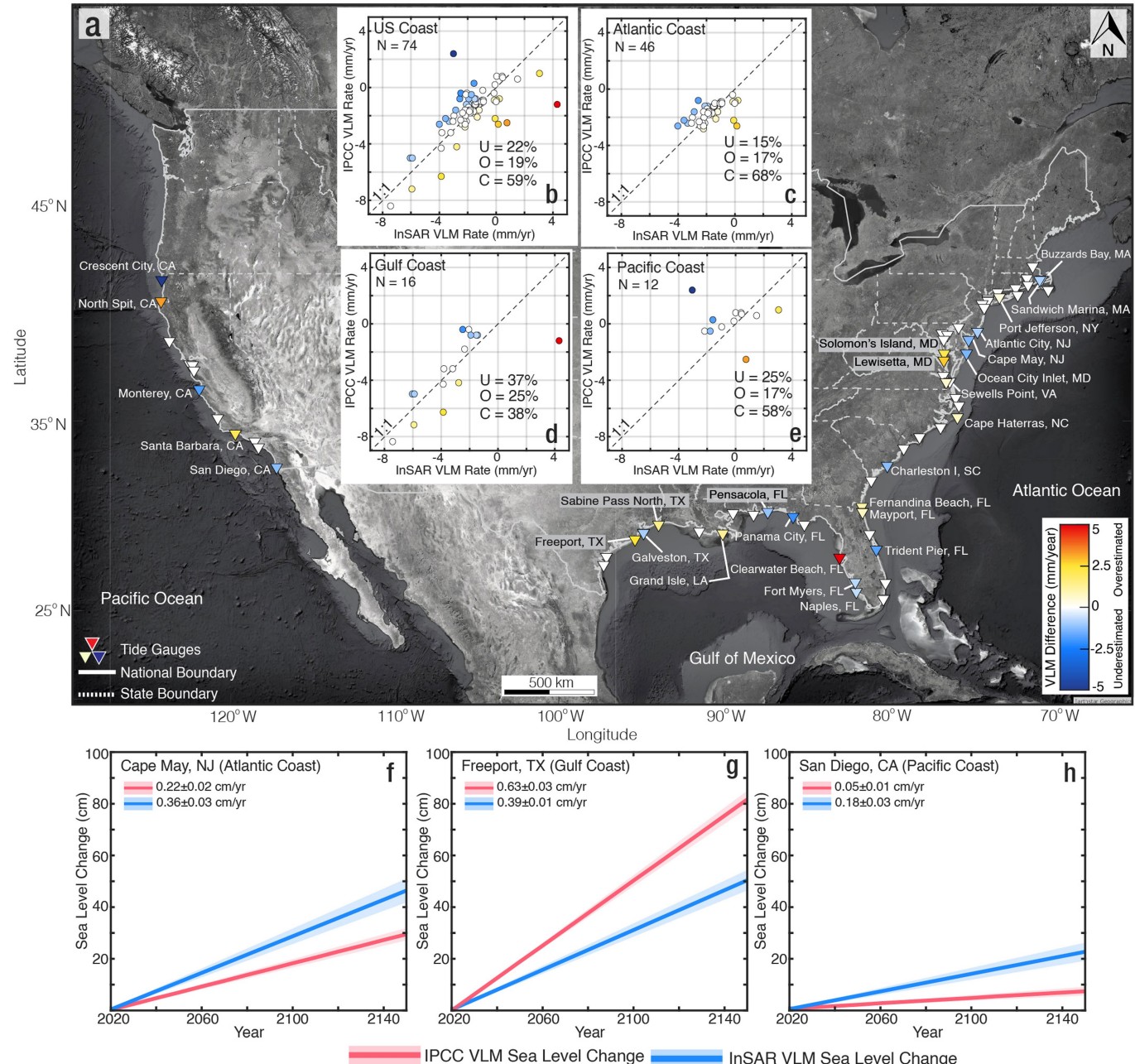

**Extended Data Fig. 6 | Sea-level change owing to VLM. a,** Difference between VLM rates from the IPCC Sixth Assessment Report[17,34] with VLM estimate from InSAR measurement at tide-gauge stations across the USA (background image: Google, Earthstar). National and state boundaries in **a** are based on public-domain vector data by the World Bank DataBank (https://data.worldbank.org/) generated in MATLAB. To obtain these rates, we averaged the VLM rates of InSAR pixels within a 200-m radius of the tide gauges and the standard deviation associated with each InSAR pixel to estimate the error ranges. A *Z*-test was initially conducted to compare the VLM rates from both sources. Stations for which no statistical difference was observed are marked in white, denoting consistent (C) VLM values. Stations for which the IPCC subsidence rates

(negative VLM) are higher than the InSAR estimates are marked in yellow or red, indicating overestimation (O), whereas stations in which the IPCC subsidence rates are lower are marked in blue, indicating underestimation (U). Note that only stations with U or O are labelled in **a**. Summary of the VLM rates from IPCC and InSAR measurements and the *Z*-test are detailed in Supplementary Table 12. Statistical comparison of IPCC VLM rate versus InSAR VLM rate for the US coast (**b**), Atlantic coast (**c**), Gulf coast (**d**) and Pacific coast (**e**). Examples of VLM comparison for the Atlantic coast (Cape May, NJ) (**f**), Gulf coast (Freeport, TX) (**g**) and Pacific coast (San Diego, CA) (**h**). State codes: MA, Massachusetts; NY, New York; NJ, New Jersey; MD, Maryland; VA, Virginia; NC, North Carolina; SC, South Carolina; GA, Georgia; FL, Florida; TX, Texas; LA, Louisiana; CA, California.

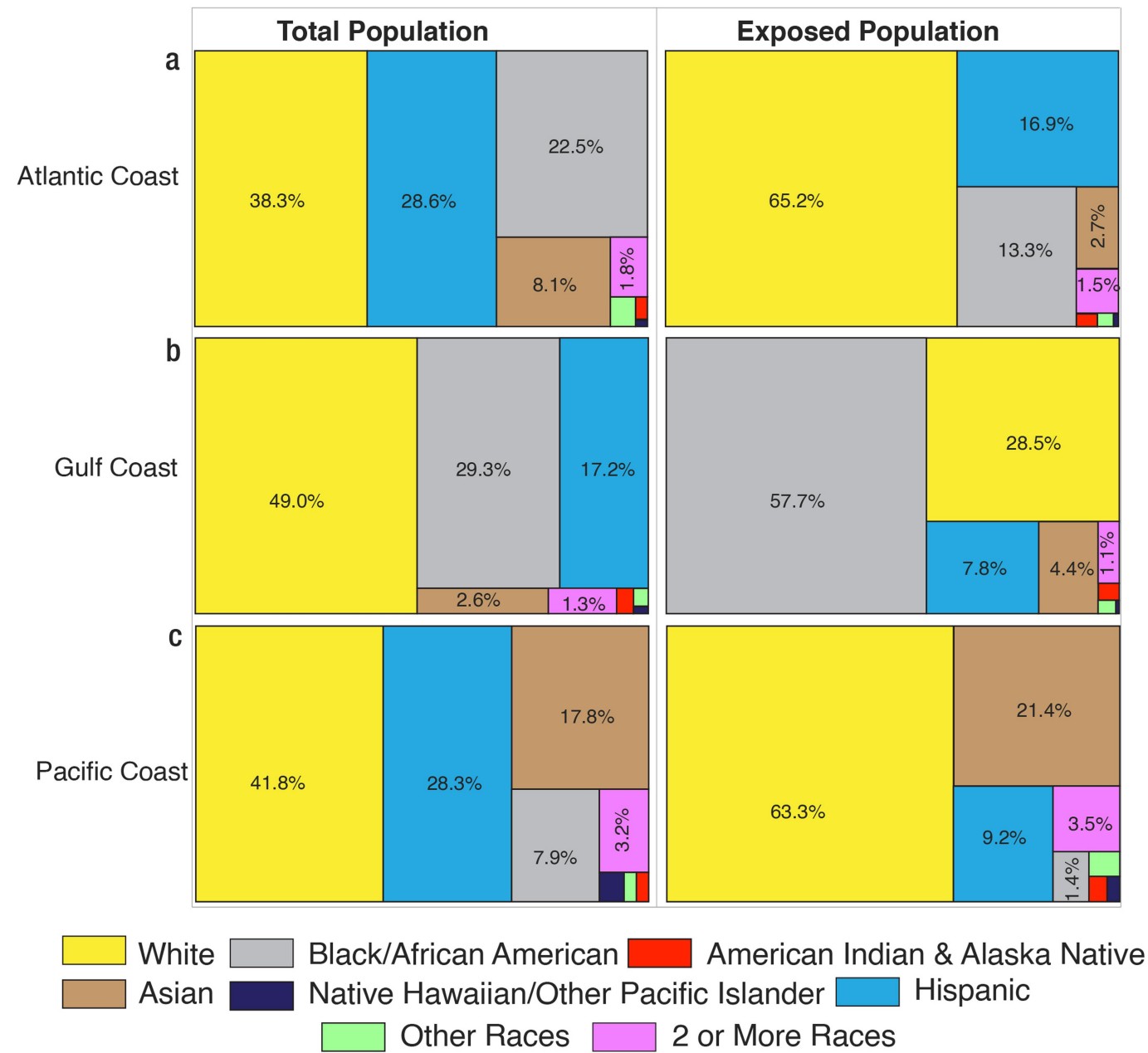

**Extended Data Fig. 7 | Total population versus exposed population by 2050 for different racial demographics on the US coast.** Tree map of total and exposed populations for the Atlantic coast (**a**), Gulf coast (**b**) and Pacific coast (**c**). The total and exposed populations are expressed as a percentage of the cumulative population for the coast. The percentages for the exposed population represent only the median exposure values evaluated using equation (3). See Supplementary Tables 13–15 for the population of each city, including the lower and upper bounds. Minoritized groups include individuals identifying as Black or African American, American Indian or Alaska Native, Asian, Native Hawaiian or Other Pacific Islander, Hispanic or Latino, other races and two or more groups.

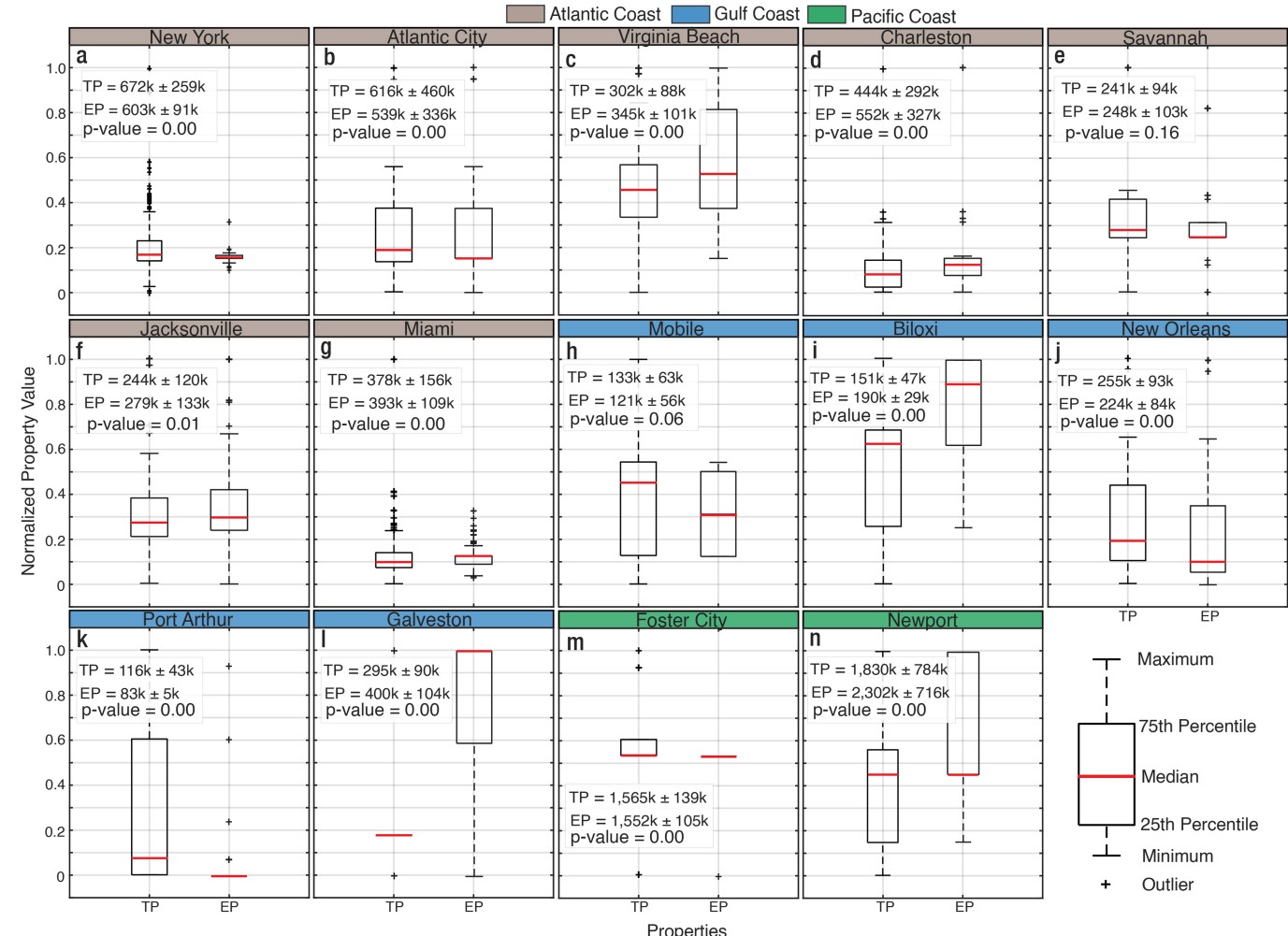

**Extended Data Fig. 8 | Modelled exposed properties value by 2050 versus the total properties value for cities on the US coast.** Box plots showing the distribution of total properties (TP) and exposure properties (EP) for New York City (**a**), Atlantic City (**b**), Virginia Beach (**c**), Charleston (**d**), Savannah (**e**), Jacksonville (**f**), Miami (**g**), Mobile (**h**), Biloxi (**i**), New Orleans (**j**), Port Arthur (**k**), Galveston (**l**), Foster City (**m**) and Newport Beach (**n**). The *y* axis for TP and EP shows the normalized values. The values for each graph show the median,

standard deviation and *P*-value for the TP and EP. The median and standard deviation values are expressed in thousands (k). For the *t*-test, the null hypothesis is that there is no statistical difference between the value of TP in the city and the value of EP. The box plots only show the median exposure values evaluated using equation (3). See Supplementary Tables 16–18 for the lower and upper bounds for each city.

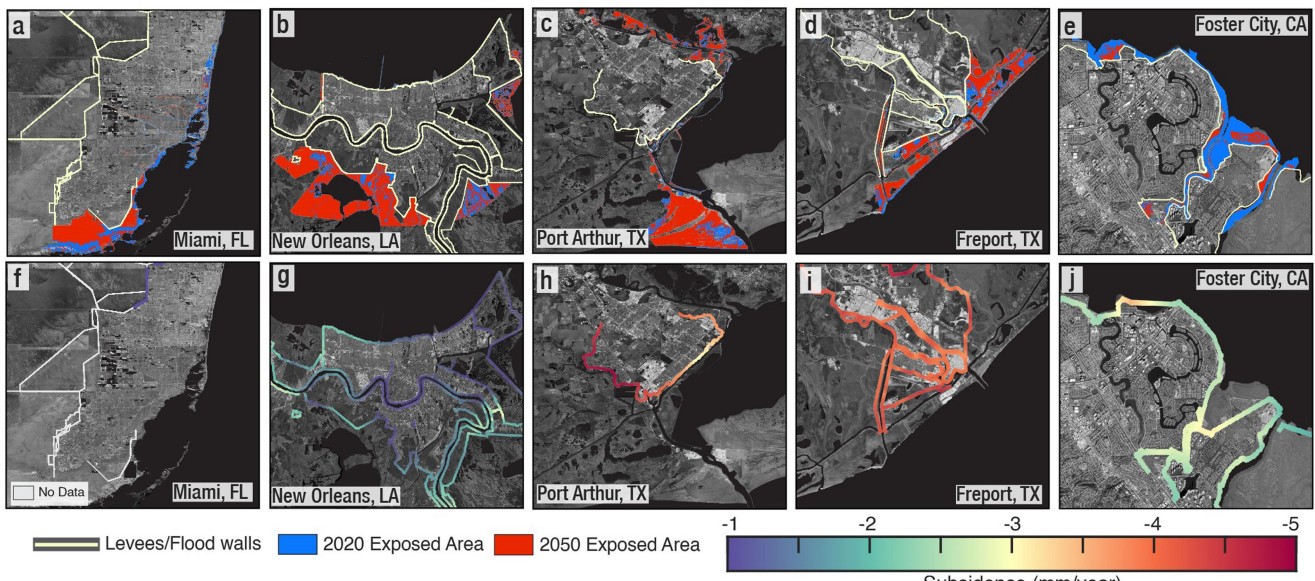

**Legend:** Levees/Flood walls | 2020 Exposed Area | 2050 Exposed Area | Subsidence (mm/year)

**Extended Data Fig. 9 | Influence of flood-control structures on modelled exposure.** Current (2020) and projected (2050) exposed areas considering defence structures for: Miami, FL (**a**); New Orleans, LA (**b**); Port Arthur, TX (**c**); Freeport, TX (**d**); Foster City, CA (**e**). The exposed areas consider VLM and SLR projection under SSP2-4.5. Exposure to subsidence for flood-control structures in: Miami, FL (**f**); New Orleans, LA (**g**); Port Arthur, TX (**h**); Freeport, TX (**i**); Foster City, CA (**j**). The yellow lines in **a**–**e** are the levees/floodwalls. Background images in **a**–**j** are from Google, Earthstar. State codes: FL, Florida; LA, Louisiana; TX, Texas; CA, California.

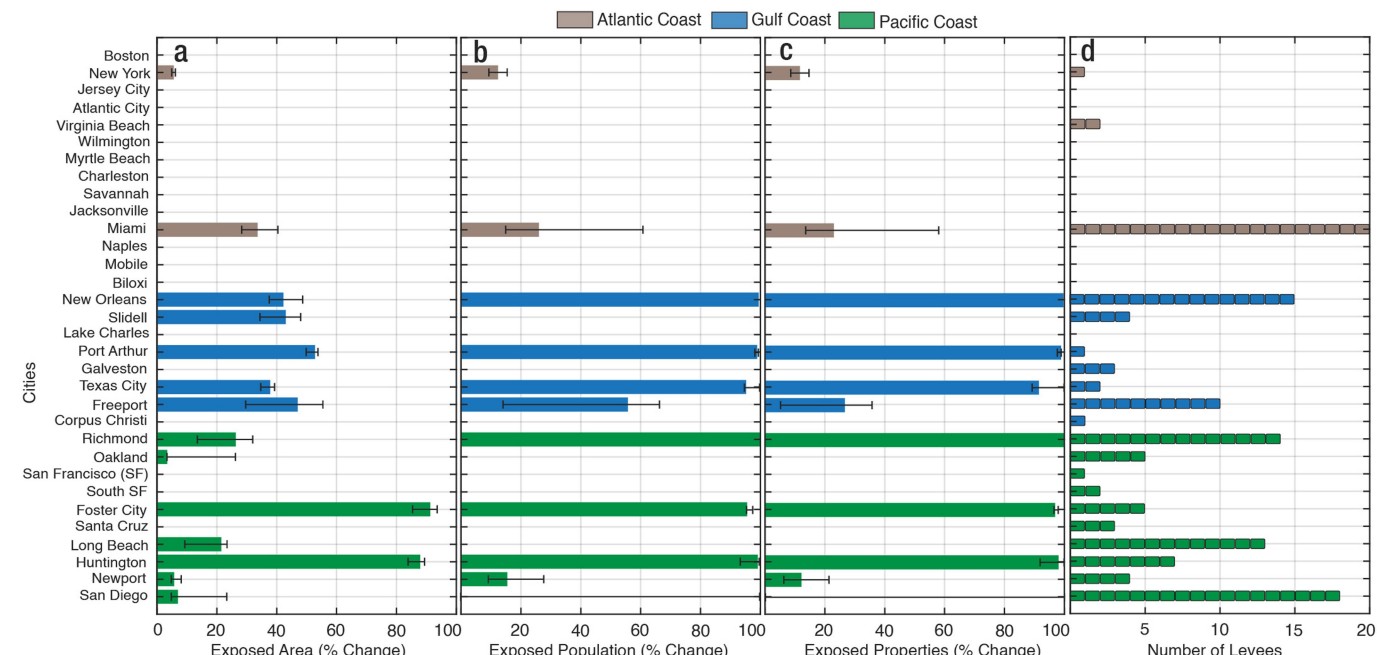

**Extended Data Fig. 10 | Influence of flood-control structures on modelled inundation.** Percent decrease in 2050 exposed area (**a**), population (**b**) and properties (**c**). **d**, Number of levees or floodwalls in each city. Extended Data Fig. 9 shows the spatial maps of the defended scenario for some selected cities. Details about the defended scenarios for each city and the percent change are shown in Supplementary Tables 20–22. The central value in **a**–**c** represents the estimated median value from equation (3), whereas the error bars represent the lower and upper bounds from equation (3).