## [Peer Review File · Nature]

Manuscript Title: Disappearing Cities on US Coasts

Reviewer Comments & Author Rebuttals

Reviewer Reports on the Initial Version:

Referees' comments:

Referee #1 (Remarks to the Author):

In this paper, the authors introduce a new compilation of InSAR-derived VLM data for US coastlines, and combine it with NOAA LiDAR DEMs, IPCC sea-level rise projections, census population and property data, Zillow home data, and Army Corps levee data to assess the 2050 exposure of current people and property.

I am not an expert on InSAR, and trust other reviewers to evaluate the quality of this data set. I focus on the remainder of the analysis.

Novelty

Here, my foremost concern is whether this analysis is novel enough to merit publication in Nature. The sorts of analyses done after the compilation of the new VLM data set are not, in and of themselves, novel -- compare, for example, Kulp, S., & Strauss, B. H. (2017). Rapid escalation of coastal flood exposure in US municipalities from sea level rise. *Climatic Change*, 142, 477-489, as well as Buchanan, M. K., Kulp, S., Cushing, L., Morello-Frosch, R., Nedwick, T., & Strauss, B. (2020). Sea level rise and coastal flooding threaten affordable housing. *Environmental Research Letters*, 15(12), 124020.

The novelty lies in the VLM data set, and thus the core question is whether exposure is "not appreciated in most US coastal cities" (as the abstract states) because previous estimates of VLM skewed exposure estimates, or because of psychological, sociological and political factors.

The core analysis missing from this paper is an assessment of whether the new VLM data significantly alters the metrics of exposure the authors use. I don't have a prior on this -- it seems equally plausible to me that (1) the InSAR VLM data are significantly different from prior VLM estimates, or (2) the InSAR VLM data may reveal locally important spatial heterogeneities but at the aggregated scale the paper focuses on, these heterogeneities do not significantly affect total population or property exposure.

To demonstrate whether the new VLM data significantly affects exposure estimates, the authors should repeat their methodology using old VLM data. The easiest way to do this would be to compare the results they would get if they used IPCC RSL projections including VLM to the ones they currently present. If the answer is that existing 'standard' VLM estimates grossly misestimate aggregate risk, great; if not, the data set is still extremely valuable at a more disaggregated scale, and I might shift the presentation to emphasize this.

One of the challenges with using 17 years of InSAR data to project 30 years into the future, as opposed to the IPCC approach of using ~100 years of tide gauge data, is the potential for aliasing short-term variability into long-term trends. The authors should give some attention to this concern when doing the comparison of the two approaches.

Uncertainty quantification

While the authors present ranges for exposure, it is unclear what uncertainties are included in these ranges. It is clear these ranges include the 17th-83rd percentile spread of the IPCC SLR projection, but I am unclear as to whether VLM uncertainties are also included in these ranges. If not, they should be; and the comparison of spread should also be part of the comparison to the IPCC projections with the IPCC's tide-gauge based VLM estimates. (I expect the uncertainty in the tide-gauge based term to be $\sim 5\times$ smaller on average than on the InSAR VLMs, so it would not surprise me at all if one of the biggest effects of substituting the new VLM is an expansion of range.)

A clear description of the approach to uncertainty propagation should be included in methods.

Baselines

As the authors note, the undefended approach "does not consider the presence of flood control structures". It is thus hard to interpret the numbers presented -- what if the majority of people exposed are *already* exposed by their calculation? I would suggest the authors focus on differences between projected exposure and current exposure, rather than their absolute calculation of projected exposure.

Where the authors discuss baselines (p. 6), some of the language is a bit hard to parse. "Current sea-level change" needs a baseline -- I assume relative to 1995-2014. I am also unclear as to the need for both baselines -- I think if the authors pick either 2020 or 1995-2014 values as the baseline, it would make the discussion clearer.

Inequality analysis

Measuring inequality only by race presents just one lens on the topic. Given that the authors are already working with home value data, I would encourage them also to look at economic inequality, perhaps treating median home value in inundated regions vs. regions as a whole as a metric. My guess is that median home value will be higher in inundated regions than in the cities as a whole on the Atlantic and Pacific coast cities, but perhaps not in Gulf coast cities.

The authors note that the Atlantic and Pacific exposed population is predominantly white, reflecting 'the demographic distribution of these regions', but based on my inspection of their results, white people seem to be disproportionately represented in the inundated regions along these coasts.

I also do not think 'underrepresented' is the right term to use in this analysis. "Underrepresented" is contextually dependent (e.g., underrepresented in STEM) -- in this case, the point of the analysis is that, along the Gulf Coast, minoritized individuals are *overrepresented* in the inundated area. (Based on examining the Table 2, I also question the value of lumping minoritized individuals together -- the major effect clearly has to do with the Black and white populations (and, along the Atlantic and especially Pacific Coast, "Other Races"). Please also note that "Other Races" is not clearly defined -- based on the text I am guessing this is primarily "Hispanic or Latino" individuals not identifying in another category.

Home valuation

I did not catch until the third reading, but the authors represent each city by a single average home value. This is a challenge to my suggestion that the authors look at economic inequality, but also limits all the economic calculations. I would encourage the authors to redo using zip-code-level ZHVI.

Other comments

page 2, line 7-8, line 25-27: It is almost certainly *not* the case that SLR will have the largest

21st century socioeconomic impact. Heat stress-related impacts (e.g., on human health and labor productivity) are almost certainly of greater socioeconomic significance because they affect a broader swath of the world's population. See, for example, Hsiang et al. (2017) (ref. 25), which shows US coastal damage at 3°C warming in 2090 on the order of 0.1-0.2% of US GDP, vs about 2% of GDP for all sectors evaluated.

page 2, line 8-11: references here are dated. Update to IPCC AR6.

page 2, line 18-22: Please confirm that all rates of SLR (global, US, city-specific) refer to the same 1920-2020 time period.

page 2, line 33: Note that ref. 28 (Ezer and Atkinson, 2014) describes a 'hot spot' of sea-level rise that has subsequently moved from the US northeast to the US southeast. For example: Ezer, T. (2019). Regional Differences in Sea Level Rise Between the Mid-Atlantic Bight and the South Atlantic Bight: Is the Gulf Stream to Blame?. *Earth's Future*, 7(7), 771-783.

page 3, line 4-5: Given the focus on the 2050 time frame, we would not expect much spread in sea-level projections among emissions scenarios. It's find to do the scenario analysis, but the differences among AR6 emissions scenario in 2050 is on the order of a couple cm.

page 4, line 13: The authors focus on SSP3-7. Through 2050, the IPCC AR6 WG3 assessment of current policy is that emissions are relatively flat, thus more similar to SSP2-4.5. Given the focus on 2050 and the point above (differences among emissions scenarios on this timescale are minor) this will not affect the final result much, but the discrepancy is notable.

Also, I believe SSP3-7 is described by AR6 as the 'high emissions scenario', and an explanation of 'medium confidence' vs 'low confidence' projections is needed if the parenthetical is used. (However, given the focus on 2050, the AR6 projections of 'low confidence' processes are essentially irrelevant, so the detail of medium vs. low confidence might be banished tot he methods.)

page 4, line 17-19: Be careful in language here. These are assessments of exposure of *current property* and *current people* who would be exposed given 2050 sea levels -- not projections of property and people exposed in 2050 (which would require projections of population and economic growth, etc).

page 4, line 25: How is it possible that there is no uncertainty in the reported percentages?

page 4, line 34-35: In addition to share of absolute numbers, it would be useful to know how different cities compare in terms of percent of population and property value exposed.

page 4, line 38: "Economic exposure" is a vague term. Substitute "home value exposure."

page 6, line 40: Hsiang et al. (2017), previously cited, also has a major focus on the inequality of climate change impacts within the United States.

page 9, line 11-15: This statement about the 1.5°C threshold is inconsistent with much of the literature, which shows harms increasingly at an accelerating rate with additional warming but not a sharp cliff at 1.5°C. The only reference among those cited which would seem to support this point is the McKay et al. (2022) paper, but this is their own sensationalistic framing; their actual finding is that the risk of tipping points increasing with warming, not that there is a cliff at 1.5°C: "Current global warming of ~1.1°C above pre-industrial already lies within the lower end of five CTP uncertainty ranges. Six CTPs become likely (with a further four possible) within the Paris Agreement range of 1.5 to <2°C warming, including collapse of the Greenland and West Antarctic ice sheets, die-off of low-latitude coral reefs, and widespread abrupt permafrost thaw. An

additional CTP becomes likely and another three possible at the $\sim 2.6^\circ\text{C}$ of warming expected under current policies."

Page 9, line 14-16: It is well established that emissions reduction have little effect on sea-level rise through 2050, so focusing on the minor differences among scenarios at 2050 seems tangential.

Page 9, line 19: "Metastasis" is a very odd word choice here, "expansion" would be more appropriate.

Data availability statement: VLM data for the Pacific Coast needs to be archived, not on Google Drive. VLM data for Gulf Coast needs a link.

Fig 1: Add comparison in panel e to the IPCC AR6 estimate of geological background rates (approximately, negative VLM) of RSL rise.

Fig 2-4: Using a log scale to present quantities with different orders of magnitudes minimizes the uncertainties shown. Using a linear scale and either separate out into different bar plots for different metrics, or adjust units so can be shown on a linear scale (e.g., show population and properties scaled by 10^4). Separate plots for quantities with different axes is probably clearer.

Fig 2-4: Units are not indicated for "High tide flooding". I assume this is square kilometers. What are we to make of cities where 'exposed area' is larger than 'high-tide flooding'?

Fig 2-4: Show differences from baseline -- either by making baseline exposure the zero point or by dividing bars into current and projected exposure.

Fig 2-4: Make version of the bar charts comparing results of using IPCC AR6 RSL estimates (including AR6 VLM), or add symbols representing the estimates using AR6 RSL estimates.

Fig 5: This figure also suffers from combining units with different orders of magnitude. My suggestion is that, when you re-do the bar charts in Figure 2-4, you find a way to indicate the reduction due to defense in each bar.

Table 1: See my prior concern that VLM uncertainty is not being propagated.

Table 2: Make sure caption is clear what the 'total' population referred to is -- I assume the total population of the cities examined.

Supplemental tables: Please label as Table S1, etc., to distinguish from main text tables.

Referee #2 (Remarks to the Author):

This study quantifies the compound impacts of subsidence and SLR at 32 major US coastal cities by 2050. It is estimated that a land area between 900 and 1,383 km², at least 1.3 million people, and 600,000 properties will be affected. The authors not only provide an advanced analysis of these changes, using state-of-the-art datasets, but also thoroughly discuss associated socio-economic impacts, climate change inequalities and adaptation strategies, making this a very relevant and comprehensive work.

This significant contribution to our understanding of the impacts of SLR is (to my knowledge) really the first study, which has ever quantified these impacts at such a large scale (for almost the entire US coast), at such a high spatial resolution, and with such an accuracy. This study is extremely important for policy makers and citizens, because it clearly illustrates the massive impacts of SLR and subsidence, which will occur already within a relatively short period of time (30 years).

The results are very well presented, and the study as a whole is very easily readable and detailed at the same time. However, there are a few points that could be clarified or improved (unless they are not due to my misunderstanding of some points). Given the social importance of the study and its quality, I support its publication.

One issue that is not entirely clear to me is whether the reported areas inundated, the people affected and the costs of damage are really likely to be as reported. You report that, on average, the combined effects of subsidence and SLR will result in a total sea level change of about 40 cm by 2050 for most of the locations considered in this study (as shown in Supplementary Figure 8). These values (=SLR and subsidence) are used to estimate the associated impacts. However, the mean (median) high water levels shown in Excel Table 12 are already higher than this figure (~60 cm). Does this mean that we should expect even worse impacts at any time during high tide? (The defended scenario wouldn't change this either). Perhaps I have made a logical error here, but I think it should be clarified (at least to me). I'm also wondering which mean sea level datum is used and from which period (see also comments below).

I think a key point of this paper is that this is really the first study to use such a comprehensive dataset (InSAR, state-of-the-art DEM data) to really quantify the future impacts (area/people/properties) of SLR on the US coasts. Obviously, you do not compare or validate your results with previous estimates because these data (like InSAR) have not really been available until now. You mention this in P3L1 (P=page), but then you don't really discuss this added value of your work anymore (except for P6L29-34). What would be interesting to understand is whether there are simply no comparable estimates of exposed area or damage from SLR in 2030 on the US coasts, or if there are, how your results would be different. I think there may be some estimates based on much simpler datasets (e.g. DEM or VLM) that also provide estimates of future impacts (e.g. https://coastal.climatecentral.org/map/11/-73.963/40.6912/?theme=sea_level_rise, or the DIVA model, etc.). Therefore, it may be worthwhile to at least highlight the novelty of your work and compare your results with existing results (if possible). For example, how different would these estimates (of damages) be if you'd replace your VLM estimates with those in the IPCC projections? Such differences are discussed in one of your earlier papers <https://doi.org/10.1038/s41467-023-37853-7>, but not really mentioned here.

Finally, you discuss very nicely in the methods section that not only the SLR projections but also the DEM and VLM data have errors (in the cm range for DEM and up to the order of 5 mm/year for VLM). For a 30-year projection, this could lead to a cumulative error of up to 15 cm in some locations. This could be very important for SL projections to 2050 (e.g. for the Gulf Coast) and even more important for periods beyond. If these effects are really relevant to your estimates, it might be worth discussing them in the main text as well. I think this might also affect your error bars in Fig. 2, which, as far as I understand, only include the uncertainties in the SLR projections.

Main text

Comments by line number (P=page, L=line):

P1L17: Abstract: 900 km², 600,000 properties, ... where do these numbers come from? Are they reported in the tables? I seem to have missed them in the text.

P2L6: 'and these effects will almost certainly grow into the future.' probably true, but it sounds a bit vague (could it be backed up by literature?)

P3L10: Do you explicitly focus on economies (except for the value of properties)?

P4L9: Relative Sea-level Rise – Future Socioeconomic Impact on the US Coasts:

- Is there a rough estimate how much % of the coastal US population is covered by these 32 cities? I noticed you mentioned that coastal population in cities accounts for 30% of the total

population, so I assume the considered cities here account for most of the coastal population, right? Might help to understand the total amount of affected people/area/value at the entire US coast.

- With respect to which year are these projections (until 2050) computed? 2023? If so, is the mean sea level (MSL) also estimated at this time? In the methods section you mention that the MSL is taken from the 1983-2000 period, is that true? In this case these estimates are rather conservative, correct?
- I'm also wondering if you were to repeat the same calculations to estimate the inundation area today in 2023 (with or without MHW levels), how much it would differ from the 2050 projections. In other words, would we already see an exposed area greater than zero? This would help me to understand whether these figures (for the undefended case), i.e. US\$ 488 billion in damages, are really to be expected as reported.
- Finally, as mentioned in the comment above, the MHW levels, are approximately as high as the projected overall relative sea level change until 2050. So does that mean that roughly the same damages occur during high tide today? Maybe I'm getting something wrong here.

P5L31: Pacific coast: Do you think that land subsidence in the cities under consideration could be affected by earthquakes or not?

P6L16: Given that some land is already below sea level today, how different is the current exposed area from the projected exposed area?

P6: Land Subsidence - An overlooked but critical driver of coastal hazards: it may be worthwhile to also be a little bit critical on the uncertainties in the projected subsidence rates. You show credible intervals in your projections, but never mention why they may be different for locations like Eugene vs. Panama, for instance (Supp. Fig. 8).

P6L18: I think you mean 'Assuming sea level change rates are held at their current levels (2020 in the IPCC sea-level projection)'

P6L29: 'However, some flooding projections due to sea-level changes do not consider the impacts of spatially variable land elevation changes, resulting in inaccurate projections of inundated areas, which may impact coastal communities' preparedness.' I think that's a really important point here, and I wonder if this can be better substantiated in comparison with other studies. As mentioned before, it's a bit unclear to me if there aren't any previous projections of inundated areas. If there are, it would be great if you could compare your results with these studies to highlight the differences and the importance of your work, and really quantify how 'inaccurate' they are.

I do not have the best overview of currently existing projections of inundation areas. But I would be really interested to understand better where the differences in projected inundation areas and existing projections come from. Below I have quickly compared the flood map for NY in the undefended case from your study and an online source from 'climate central'.

([https://coastal.climatecentral.org/map/11/-](https://coastal.climatecentral.org/map/11/-73.963/40.6912/?theme=sea_level_rise&map_type=year&basemap=roadmap&contiguous=false&elevation_model=best_available&forecast_year=2100&pathway=ssp3rcp70&percentile=p50&refresh=true&return_level=return_level_1&rl_model=tebaldi_2012&slr_model=ipcc_2021_med)

[73.963/40.6912/?theme=sea_level_rise&map_type=year&basemap=roadmap&contiguous=false&elevation_model=best_available&forecast_year=2100&pathway=ssp3rcp70&percentile=p50&refresh=true&return_level=return_level_1&rl_model=tebaldi_2012&slr_model=ipcc_2021_med](https://coastal.climatecentral.org/map/11/-73.963/40.6912/?theme=sea_level_rise&map_type=year&basemap=roadmap&contiguous=false&elevation_model=best_available&forecast_year=2100&pathway=ssp3rcp70&percentile=p50&refresh=true&return_level=return_level_1&rl_model=tebaldi_2012&slr_model=ipcc_2021_med)).

(Figures are in the attached PDF)

Of course there are some differences in the settings here. E.g. the projections of 'climate central (CC)' are for 2100 (vs. 2050 in your case), they may use a different DEM (<https://coast.noaa.gov/digitalcoast/data/coastallidar.html>), and they employ the VLM data from the latest projections.

There are some differences in the projected inundation areas in these maps. For example, the region north of Coney Island is not really seen as affected in the CC projections, while regions

around the Hackensack River are more likely to be affected in the CC projections.

So I'm just trying to understand, where these local differences come from. Is this due to the different VLM data used or the DEM? I think clarification of these differences, would help me (and the reader) to better understand the value and the robustness of these new estimates of projected inundated areas.

Below, I plot the subsidence rates (in cm/year) as used in your study (from <https://doi.org/10.7294/19350959>). Could it be that the region north of Coney Island is more likely to be affected (in your study compared to CC) because you estimate higher subsidence rates there?

P6L33: Isn't it rather the spatial resolution that is unprecedented? GNSS has a much higher temporal resolution.

P7L6-7: Could this inequality effect be even stronger, when extreme sea levels from storm surges, hurricanes etc. are taken into account (as you explain below)? P7L1-3: 'approximately' reflecting the demographic distribution?

P8L12, Fig. 5, and sup. Fig. 9: Again, I may have misunderstood something, but to follow up on previous comments the MHW in New York is 0.66m (Excel Table 12). Does this mean that, even in the case defended here, the area exposed at high tide today is already at least as large as the projected area exposed in 2050 (red, without high tide flooding)?

P8L23: '76 to 99% of the exposed area, population, and properties being situated on the Atlantic coast.' Does 99% refer to population and properties?

Supplementary Figures/Tables of the main File:

Fig: 2k: 'error bars represent the lower (17th percentile) and upper (83rd percentile) bounds for projected SLR'. Does that mean that the VLM uncertainty intervals, or DEM uncertainties are not included here?

Table 2: Please indicate the period of these projections, and the IPCC SLR scenario.

Fig. 5: Maybe mention the IPCC SLR scenario in the caption.

Materials and Methods:

P10L42: If the original data source is NGL, I would acknowledge that, together with Shirzaei et al. 36.

P12: High-Tide Estimates: Does that mean you use the MSL (mean sea level) datum computed over 1983-2001? If so, to estimate exposure in 2050, wouldn't you need to compute the changes starting from that period? Or, over which period do you compute the accumulated SLC and VLM effects? Over 2020-2050 or starting from the 1983-2001 period?

P12L44: 'assuming a linear VLM rate to 2050', when do you start this extrapolation?

P13L11: 'We consider a census block inundated if greater than 20% of its area is inundated'. Will assuming a different figure here (20%) massively affect the results presented? Is this assumption of how much area needs to be flooded (to be counted as flooded) linearly proportional to your final estimates of the area affected? For example, if you assume that 40% has to be flooded (i.e. twice the current assumption) for a census block to be counted, would this mean that on average only half the population/area would be affected?

Why is 20% a conservative estimate? Is it based on the distribution of street and building heights, etc.?

P10L10: Maybe check again if this link works? It didn't work for me.

Supplementary File:

Fig. 7: In the caption (of a, b, and c) it is not clarified that the period over which the changes will occur is 2020-2050 (that's only done in the main text) by assuming constant trends over this period

Fig. 9. What scenario is used here?

(Please find my complete review in the PDF file)

Author Rebuttals to Initial Comments:

Reviewer #1 (Remarks to the Author): (Reviewers comments in normal text, response to reviewers are in bold)

In this paper, the authors introduce a new compilation of InSAR-derived VLM data for US coastlines, and combine it with NOAA LiDAR DEMs, IPCC sea-level rise projections, census population and property data, Zillow home data, and Army Corps levee data to assess the 2050 exposure of current people and property.

I am not an expert on InSAR, and trust other reviewers to evaluate the quality of this data set. I focus on the remainder of the analysis.

We appreciate this reviewer's time and efforts in evaluating the manuscript. Your expert comments and suggestions were critical in identifying the key areas where our work can be improved, refined, and clarified. Key changes made to the manuscript following your recommendation include:

- 1. Differences between InSAR VLM and IPCC VLM**
- 2. Differences in the exposure from this study and IPCC RSL**
- 3. Propagation of all uncertainties: VLM, DEM and sea level change**
- 4. Reporting additional projected exposure, with reference to the current as opposed to absolute values.**
- 5. Utilizing the zip code Zillow home value index for home value exposure.**
- 6. Calculating the inequality in the home value exposure in addition to the inequality in exposed population.**

These highlighted changes are discussed in detail below:

Novelty

Here, my foremost concern is whether this analysis is novel enough to merit publication in Nature. The sorts of analyses done after the compilation of the new VLM data set are not, in and of themselves, novel -- compare, for example, Kulp, S., & Strauss, B. H. (2017). Rapid escalation of coastal flood exposure in US municipalities from sea level rise. *Climatic Change*, 142, 477-489, as well as Buchanan, M. K., Kulp, S., Cushing, L., Morello-Frosch, R., Nedwick, T., & Strauss, B. (2020). Sea level rise and coastal flooding threaten affordable housing. *Environmental Research Letters*, 15(12), 124020.

The novelty lies in the VLM data set, and thus the core question is whether exposure is "not appreciated in most US coastal cities" (as the abstract states) because previous estimates of VLM skewed exposure estimates, or because of psychological, sociological and political factors.

The core analysis missing from this paper is an assessment of whether the new VLM data significantly alters the metrics of exposure the authors use. I don't have a prior on this – it seems equally plausible to me that (1) the InSAR VLM data are significantly different from prior VLM estimates, or (2) the InSAR VLM data may reveal locally important spatial heterogeneities but at the aggregated scale the paper focuses on, these heterogeneities do not significantly affect total population or property exposure.

To demonstrate whether the new VLM data significantly affects exposure estimates, the authors should repeat their methodology using old VLM data. The easiest way to do this would be to

compare the results they would get if they used IPCC RSL projections including VLM to the ones they currently present. If the answer is that existing 'standard' VLM estimates grossly misestimate aggregate risk, great; if not, the data set is still extremely valuable at a more disaggregated scale, and I might shift the presentation to emphasize this.

We thank the reviewer for their comments. We agree that the novelty of our study lies in the new Vertical Land Motion (VLM) dataset. The novel VLM datasets combined with the rest of our analysis result in maps and models that are unique in terms of resolution and spatial extent. To explicitly demonstrate the VLM impact on exposure estimates, we have conducted a comparative analysis using the VLM estimates (primarily from tide gauges) employed in IPCC Relative Sea Level (RSL) projections (Figure 5). The results show significant deviations between the exposure in this study and the exposure when using the IPCC RSLR data, these differences, as highlighted in Figure 5d result primarily from differences between IPCC VLM at tide gauges and InSAR VLM for the cities. This is because VLM occurring at tide gauges often do not represent the VLM within the cities. Additionally, we compared our InSAR VLM with the IPCC VLM at the location of the tide gauges and showed differences between these rates. While the VLM from tide gauges employs longer time intervals (~100 years). It is unlikely that the sinking rates from the past 70 – 100 years would continue into the future. This is particularly important where anthropogenic activities drive subsidence. Further, tide gauge VLM often misses signals due to shallow compaction processes. We have discussed and emphasized this point on page 7 lines 6 – 27 of the manuscript.

One of the challenges with using 17 years of InSAR data to project 30 years into the future, as opposed to the IPCC approach of using ~100 years of tide gauge data, is the potential for aliasing short-term variability into long-term trends. The authors should give some attention to this concern when doing the comparison of the two approaches.

We thank the reviewer for their comments. This is likely a limitation of every effort projection of VLM, as the driving mechanism can vary in time and space. We have discussed the issue in our prior publication (Shirzaei et al. 2021 Nature Rev) and actively work to minimize the impact of short-term signal transients on long-term projections. Notably, the InSAR VLM data utilized in this study was created by performing a joint inversion of Sentinel-1, ALOS, and GNSS datasets. Inherent in the decomposition of the analysis is the removal of short-term variability from the SAR line-of-sight datasets and GNSS data. Thus, annual or interannual variabilities are removed from the datasets. Subsequently, we also performed validation using long-term GNSS data. The comparative analysis revealed a correlation between the GNSS and the InSAR data as shown on supplementary figure 15.

Uncertainty quantification

While the authors present ranges for exposure, it is unclear what uncertainties are included in these ranges. It is clear these ranges include the 17th-83rd percentile spread of the IPCC SLR projection, but I am unclear as to whether VLM uncertainties are also included in these ranges. If not, they should be; and the comparison of spread should also be part of the comparison to

the IPCC projections with the IPCC's tide-gauge based VLM estimates. (I expect the uncertainty in the tide-gauge based term to be ~5x smaller on average than on the InSAR VLMs, so it would not surprise me at all if one of the biggest effects of substituting the new VLM is an expansion of range.)

A clear description of the approach to uncertainty propagation should be included in methods.

We thank the reviewer for their comment, this is really important. We have now propagated all errors in the subsequent exposure analysis, this include the DEM error, VLM uncertainties (1 standard deviation) and IPCC SLR (17th and 83rd percentile). A description of the uncertainty propagation has been included in the methods section on pages 14, line 45 to page 15, line 12.

Baselines

As the authors note, the undefended approach "does not consider the presence of flood control structures". It is thus hard to interpret the numbers presented -- what if the majority of people exposed are *already* exposed by their calculation? I would suggest the authors focus on differences between projected exposure and current exposure, rather than their absolute calculation of projected exposure.

We sincerely appreciate the reviewer's suggestions. We have now updated the figures, text, and tables to reflect additional exposed area by 2050 compared to the baseline of 2020. 2050 exposure refers to the additional areas/population/property exposure to high tide flooding by 2050 (see page 14, lines 19 to 31; supplementary tables 2 to 4).

Where the authors discuss baselines (p. 6), some of the language is a bit hard to parse. "Current sea-level change" needs a baseline -- I assume relative to 1995-2014. I am also unclear as to the need for both baselines -- I think if the authors pick either 2020 or 1995-2014 values as the baseline, it would make the discussion clearer.

We thank the reviewer for their excellent suggestion. Current sea-level change refers to areas not exposed to high tide, then using 2020 as a baseline, we estimate additional areas exposed considering relative SLR. This has been clarified in the manuscript on page 14, lines 23 to 43.

Inequality analysis

Measuring inequality only by race presents just one lens on the topic. Given that the authors are already working with home value data, I would encourage them also to look at economic inequality, perhaps treating median home value in inundated regions vs. regions as a whole as a metric. My guess is that median home value will be higher in inundated regions than in the cities as a whole on the Atlantic and Pacific coast cities, but perhaps not in Gulf coast cities.

We thank the reviewer for this suggestion. Measuring inequality from multiple facets is an important part of measuring inequality. We have included additional supplementary tables 15 to 17 and supplementary figure 11, which examines economic inequality using the home value dataset. For this analysis, we utilize a t-test for comparing the total home value versus exposed home value. Indeed, we find that in some cities on the Gulf and

Atlantic coasts, the median home value in the inundated areas are less than the median home value in the city. We sincerely appreciate the suggestion.

The authors note that the Atlantic and Pacific exposed population is predominantly white, reflecting 'the demographic distribution of these regions', but based on my inspection of their results, white people seem to be disproportionately represented in the inundated regions along these coasts.

Yes, this is correct, we have rephrased this section for clarity to present this. However, it is important to recognize that on the Atlantic coast and Pacific coasts, the White population are the most populous total population on these coasts and remains the most populous exposed population. However, on the Gulf Coast, black population are not the most populous total population but are the most exposed population. This point is important, particularly for a minority group.

I also do not think 'underrepresented' is the right term to use in this analysis.

"Underrepresented" is contextually depend (e.g., underrepresented in STEM) -- in this case, the point of the analysis is that, along the Gulf Coast, minoritized individuals are *overrepresented* in the inundated area. (Based on examining the Table 2, I also question the value of lumping minoritized individuals together -- the major effect clearly has to do with the Black and white populations (and, along the Atlantic and especially Pacific Coast, "Other Races"). Please also note that "Other Races" is not clearly defined -- based on the text I am guessing this is primarily "Hispanic or Latino" individuals not identifying in another category.

We thank the reviewer for this comment. Yes, the minority should be the right term. This has been changed throughout the manuscript. Also, we have modified the language to reflect the use of overrepresented in the different coasts. We now discuss the different groups separately. We have also included Hispanic or Latino population.

Home valuation

I did not catch until the third reading, but the authors represent each city by a single average home value. This is a challenge to my suggestion that the authors look at economic inequality, but also limits all the economic calculations. I would encourage the authors to redo using zip-code-level ZHVI.

We have now repeated the analysis using the zip-code ZHVI. We thank the reviewer for this suggestion.

Other comments

page 2, line 7-8, line 25-27: It is almost certainly *not* the case that SLR will have the largest 21st century socioeconomic impact. Heat stress-related impacts (e.g., on human health and labor productivity) are almost certainly of greater socioeconomic significance because they affect a broader swath of the world's population. See, for example, Hsiang et al. (2017) (ref. 25), which shows US coastal damage at 3°C warming in 2090 on the order of 0.1-0.2% of US GDP, vs about 2% of GDP for all sectors evaluated.

We thank the reviewer for this suggestion. This has been updated.

page 2, line 8-11: references here are dated. Update to IPCC AR6.

Citation has been updated.

page 2, line 18-22: Please confirm that all rates of SLR (global, US, city-specific) refer to the same 1920-2020 time period.

Yes, that is the case.

page 2, line 33: Note that ref. 28 (Ezer and Atkinson, 2014) describes a 'hot spot' of sea-level rise that has subsequently moved from the US northeast to the US southeast. For example: Ezer, T. (2019). Regional Differences in Sea Level Rise Between the Mid-Atlantic Bight and the South Atlantic Bight: Is the Gulf Stream to Blame?. *Earth's Future*, 7(7), 771-783.

Noted.

page 3, line 4-5: Given the focus on the 2050 time frame, we would not expect much spread in sea-level projections among emissions scenarios. It's find to do the scenario analysis, but the differences among AR6 emissions scenario in 2050 is on the order of a couple cm.

Yes, that is the case. This is highlighted on page 2, lines 37-43.

page 4, line 13: The authors focus on SSP3-7. Through 2050, the IPCC AR6 WG3 assessment of current policy is that emissions are relatively flat, thus more similar to SSP2-4.5. Given the focus on 2050 and the point above (differences among emissions scenarios on this timescale are minor) this will not affect the final result much, but the discrepancy is notable. Also, I believe SSP3-7 is described by AR6 as the 'high emissions scenario', and an explanation of 'medium confidence' vs 'low confidence' projections is needed if the parenthetical is used. (However, given the focus on 2050, the AR6 projections of 'low confidence' processes are essentially irrelevant, so the detail of medium vs. low confidence might be banished tot he methods.)

We thank the reviewer for this observation. We have modified our analysis to utilize the SSP 2-4.5, the current trajectory from the IPCC AR6 report. We have removed the reference to medium scenario throughout the manuscript.

page 4, line 17-19: Be careful in language here. These are assessments of exposure of *current property* and *current people* who would be exposed given 2050 sea levels -- not projections of property and people exposed in 2050 (which would require projections of population and economic growth, etc).

We thank the reviewer for the comments, we have adjusted the text accordingly.

page 4, line 25: How is it possible that there is no uncertainty in the reported percentages?

This has been updated throughout the text. Thanks!

page 4, line 34-35: In addition to share of absolute numbers, it would be useful to know how different cities compare in terms of percent of population and property value exposed.

We have included some text about this for the different coasts.

page 4, line 38: "Economic exposure" is a vague term. Substitute "home value exposure."

This has been modified throughout the manuscript.

page 6, line 40: Hsiang et al. (2017), previously cited, also has a major focus on the inequality of climate change impacts within the United States.

This citation has been included in this sentence.

page 9, line 11-15: This statement about the 1.5°C threshold is inconsistent with much of the literature, which shows harms increasingly at an accelerating rate with additional warming but not a sharp cliff at 1.5°C. The only reference among those cited which would seem to support this point is the McKay et al. (2022) paper, but this is their own sensationalistic framing; their actual finding is that the risk of tipping points increasing with warming, not that there is a cliff at 1.5°C: "Current global warming of ~1.1°C above pre-industrial already lies within the lower end of five CTP uncertainty ranges. Six CTPs become likely (with a further four possible) within the Paris Agreement range of 1.5 to <2°C warming, including collapse of the Greenland and West Antarctic ice sheets, die-off of low-latitude coral reefs, and widespread abrupt permafrost thaw. An additional CTP becomes likely and another three possible at the ~2.6°C of warming expected under current policies."

We have modified the text in line with the comments. We thank the reviewer for this comment.

Page 9, line 14-16: It is well established that emissions reduction have little effect on sea-level rise through 2050, so focusing on the minor differences among scenarios at 2050 seems tangential.

We thank the reviewer for this observation. We have modified the text accordingly.

Page 9, line 19: "Metastasis" is a very odd word choice here, "expansion" would be more appropriate.

Modified to expansion.

Data availability statement: VLM data for the Pacific Coast needs to be archived, not on Google Drive. VLM data for Gulf Coast needs a link.

The link for the Gulf Coast has been included, also we have moved the Pacific coast to a repository.

Fig 1: Add comparison in panel e to the IPCC AR6 estimate of geological background rates (approximately, negative VLM) of RSL rise.

We included this in figure 5d. As a separate plot.

Fig 2-4: Using a log scale to present quantities with different orders of magnitudes minimizes the uncertainties shown. Using a linear scale and either separate out into different bar plots for different metrics, or adjust units so can be shown on a linear scale (e.g., show population and properties scaled by 10⁴). Separate plots for quantities with different axes is probably clearer.

We have modified this and now display all the result using a linear scale.

Fig 2-4: Units are not indicated for "High tide flooding". I assume this is square kilometers. What are we to make of cities where 'exposed area' is larger than 'high-tide flooding'?

This has been included in the figure. Thank you for the observation. The new analysis, now focuses on current areas exposed to high tide flooding and additional areas exposed to high-tide flooding by 2050 due to relative SLR.

Fig 2-4: Show differences from baseline -- either by making baseline exposure the zero point or by dividing bars into current and projected exposure.

The baseline exposure is made at zero and we report the additional exposed area and properties by 2050. Current exposure is highlighted in supplementary tables 2 to 4.

Fig 2-4: Make version of the bar charts comparing results of using IPCC AR6 RSL estimates (including AR6 VLM), or add symbols representing the estimates using AR6 RSL estimates.

This has been included as figure 5. We thank the reviewer for the suggestion.

Fig 5: This figure also suffers from combining units with different orders of magnitude. My suggestion is that, when you re-do the bar charts in Figure 2-4, you find a way to indicate the reduction due to defense in each bar.

We thank the reviewer for their comments. This has been updated.

Table 1: See my prior concern that VLM uncertainty is not being propagated.

We thank the reviewer for their comment. VLM and DEM errors are now propagated.

Table 2: Make sure caption is clear what the 'total' population referred to is -- I assume the total population of the cities examined.

This table has been removed and replaced with a figure. But yes, the total population refers to the total population from the cities examined.

Supplemental tables: Please label as Table S1, etc., to distinguish from main text tables.

Noted! This has been updated.

Reviewer #2 (Remarks to the Author): (Reviewers comments in normal text, response to reviewers are in bold)

1. Comments for Author

This study quantifies the compound impacts of subsidence and SLR at 32 major US coastal cities by 2050. It is estimated that a land area between 900 and 1,383 km², at least 1.3 million people, and 600,000 properties will be affected. The authors not only provide an advanced analysis of these changes, using state-of-the-art datasets, but also thoroughly discuss associated socio-economic impacts, climate change inequalities and adaptation strategies, making this a very relevant and comprehensive work. This significant contribution to our understanding of the impacts of SLR is (to my knowledge) really the first study, which has ever quantified these impacts at such a large scale (for almost the entire US coast), at such a high spatial resolution, and with such an accuracy. This study is extremely important for policy makers and citizens, because it clearly illustrates the massive impacts of SLR and subsidence, which will occur already within a relatively short period of time (30 years). The results are very well presented, and the study as a whole is very easily readable and detailed at the same time. However, there are a few points that could be clarified or improved (unless they are not due to my misunderstanding of some points). Given the social importance of the study and its quality, I support its publication.

We are thankful to the reviewer for their comprehensive reviews of our manuscript. The insights, comments, and questions provided have been invaluable in helping us enhance this research and manuscript.

Main Comments

One issue that is not entirely clear to me is whether the reported areas inundated, the people affected and the costs of damage are really likely to be as reported. You report that, on average, the combined effects of subsidence and SLR will result in a total sea level change of about 40 cm by 2050 for most of the locations considered in this study (as shown in Supplementary Figure 8). These values (=SLR and subsidence) are used to estimate the associated impacts. However, the mean (median) high water levels shown in Excel Table 12 are already higher than this figure (~60 cm). Does this mean that we should expect even worse impacts at any time during high tide? (The defended scenario wouldn't change this either). Perhaps I have made a logical error here, but I think it should be clarified (at least to me). I'm also wondering which mean sea level datum is used and from which period (see also comments below).

We thank the reviewer for this important observation. To avoid any ambiguity and in line with this and other suggested comments (including those of reviewer 1), we have refocused the projected scenarios to reflect the additional areas that would be affected by high tide events (MHW) by 2050 compared to the current exposure. In the updated manuscript, we first report the current (2020) MHW-exposed areas, population, and properties. The 2050 exposure are the additional exposed areas, population, and properties that would be exposed to the MHW considering Relative SLR (VLM and SLR) by 2050. The sea level change datum used here, thus uses 2020 as a baseline. In the section: "Land Subsidence - An overlooked but critical driver of coastal hazards," we report the areas affected by just land subsidence and the combination of land subsidence and SLR.

I think a key point of this paper is that this is really the first study to use such a comprehensive dataset (InSAR, state-of-the-art DEM data) to really quantify the future impacts (area/people/properties) of SLR on the US coasts. Obviously, you do not compare or validate your results with previous estimates because these data (like InSAR) have not really been available until now. You mention this in P3L1 (P=page), but then you don't really discuss this added value of your work anymore (except for P6L29-34). What would be interesting to understand is whether there are simply no comparable estimates of exposed area or damage from SLR in 2030 on the US coasts, or if there are, how your results would be different. I think there may be some estimates based on much simpler datasets (e.g. DEM or VLM) that also provide estimates of future impacts (e.g. https://coastal.climatecentral.org/map/11/-73.963/40.6912/?theme=sea_level_rise, or the DIVA model, etc.). Therefore, it may be worthwhile to at least highlight the novelty of your work and compare your results with existing results (if possible). For example, how different would these estimates (of damages) be if you'd replace your VLM estimates with those in the IPCC projections? Such differences are discussed in one of your earlier papers <https://doi.org/10.1038/s41467-023-37853-7>, but not really mentioned here. Finally, you discuss very nicely in the methods section that not only the SLR projections but also the DEM and VLM data have errors (in the cm range for DEM and up to the order of 5 mm/year for VLM). For a 30-year projection, this could lead to a cumulative error of up to 15 cm in some locations. This could be very important for SL projections to 2050 (e.g. for the Gulf Coast) and even more important for periods beyond. If these effects are really relevant to your estimates, it might be worth discussing them in the main text as well. I think this might also affect your error bars in Fig. 2, which, as far as I understand, only include the uncertainties in the SLR projections.

We thank the reviewer for their suggestion. To make such a comparison and determine if differences are as a result of the VLM, we have to keep all other variables constant (DEM, and SLR), while varying the VLM. To that end, we have now included a comparative analysis of our study's exposure against the RSLR from the IPCC scenarios. Also, a comparison of the VLM estimates (see figure 5). We also compared the VLM at IPCC tide gauges with our VLM estimates. This added value of our dataset is discussed on page 7, lines 6 to 28. Additionally, in the new analysis, we have projected the errors from the DEM and InSAR VLM, with the uncertainties from the IPCC sea level change to determine the error bars (lower and upper bounds) of exposure. We really appreciate this valuable insight. We sincerely thank the reviewer for their excellent suggestion.

Main text

Comments by line number (P=page, L=line):

P1L17: Abstract: 900 km², 600,000 properties, ... where do these numbers come from? Are they reported in the tables? I seem to have missed them in the text.

This number comes from the defended scenario, which is discussed in the section "towards sustainable adaptation strategies."

P2L6: 'and these effects will almost certainly grow into the future.' probably true, but it sounds a bit vague (could it be backed up by literature?)

A reference for this has been included - the IPCC report.

P3L10: Do you explicitly focus on economies (except for the value of properties)?

We thank the reviewer for their suggestion. We have deleted the economies from this sentence.

P4L9: Relative Sea-level Rise – Future Socioeconomic Impact on the US Coasts: Is there a rough estimate how much % of the coastal US population is covered by these 32 cities? I noticed you mentioned that coastal population in cities accounts for 30% of the total population, so I assume the considered cities here account for most of the coastal population, right? Might help to understand the total amount of affected people/area/value at the entire US coast.

We thank the reviewer for this comment. This has been included on page 3; lines 27-28, where we first reference the 32 coastal cities. Also, we included supplementary table 1, which shows the total population, properties, and home value for each city analyzed (see page 4, lines 14 and 15).

With respect to which year are these projections (until 2050) computed? 2023? If so, is the mean sea level (MSL) also estimated at this time? In the methods section you mention that the MSL is taken from the 1983-2000 period, is that true? In this case these estimates are rather conservative, correct?

The 2050 projections are now made using 2020 as the baseline. And the MHW uses NAVD88 as the datum, corresponding to the datum of the DEM data.

I'm also wondering if you were to repeat the same calculations to estimate the inundation area today in 2023 (with or without MHW levels), how much it would differ from the 2050 projections. In other words, would we already see an exposed area greater than zero? This would help me to understand whether these figures (for the undefended case), i.e. US\$ 488 billion in damages, are really to be expected as reported. Finally, as mentioned in the comment above, the MHW levels, are approximately as high as the projected overall relative sea level change until 2050. So does that mean that roughly the same damages occur during high tide today? Maybe I'm getting something wrong here.

Yes, that is the case. To account for this and avoid any ambiguities, we now report additional exposure, rather than absolute values. This has been updated throughout the manuscript and tables.

P5L31: Pacific coast: Do you think that land subsidence in the cities under consideration could be affected by earthquakes or not?

Certainly, we cannot rule out completely the possibility of an earthquake in these Pacific cities.

P6L16: Given that some land is already below sea level today, how different is the current exposed area from the projected exposed area?

We thank the reviewer for their comment. We have updated the manuscript to only highlight differences between current versus projected exposure.

P6: Land Subsidence - An overlooked but critical driver of coastal hazards: it may be worthwhile to also be a little bit critical on the uncertainties in the projected subsidence rates. You show

credible intervals in your projections, but never mention why they may be different for locations like Eugene vs. Panama, for instance (Supp. Fig. 8).

The uncertainties in the VLM projections arise from the standard deviation in the VLM motion data. We highlight this in the methods section. Where we discuss the lower precision (i.e. higher standard deviation values along some coasts) in detail on page 12; lines 12 to 30.

P6L18: I think you mean 'Assuming sea level change rates are held at their current levels (2020 in the IPCC sea-level projection)'

Yes, that is the case. However, this sentence has been removed from the updated manuscript.

P6L29: 'However, some flooding projections due to sea-level changes do not consider the impacts of spatially variable land elevation changes, resulting in inaccurate projections of inundated areas, which may impact coastal communities' preparedness.' I think that's a really important point here, and I wonder if this can be better substantiated in comparison with other studies. As mentioned before, it's a bit unclear to me if there aren't any previous projections of inundated areas. If there are, it would be great if you could compare your results with these studies to highlight the differences and the importance of your work, and really quantify how 'inaccurate' they are. I do not have the best overview of currently existing projections of inundation areas. But I would be really interested to understand better where the differences in projected inundation areas and existing projections come from. Below I have quickly compared the flood map for NY in the undefended case from your study and an online source from 'climate central'.

(https://coastal.climatecentral.org/map/11/73.963/40.6912/?theme=sea_level_rise&map_type=year&basemap=roadmap&contiguous=false&elevation_model=best_available&forecast_year=2100&pathway=ssp3rcp70&percentile=p50&refresh=true&return_level=return_level_1&rl_model=ebaldi_2012&slr_model=ipcc_2021_med). Of course there are some differences in the settings here. E.g. the projections of 'climate central (CC)' are for 2100 (vs. 2050 in your case), they may use a different DEM (<https://coast.noaa.gov/digitalcoast/data/coastallidar.html>), and they employ the VLM data from the latest projections.

There are some differences in the projected inundation areas in these maps. For example, the region north of Coney Island is not really seen as affected in the CC projections, while regions around the Hackensack River are more likely to be affected in the CC projections. So I'm just trying to understand, where these local differences come from. Is this due to the different VLM data used or the DEM? I think clarification of these differences, would help me (and the reader) to better understand the value and the robustness of these new estimates of projected inundated areas.

Below, I plot the subsidence rates (in cm/year) as used in your study (from <https://doi.org/10.7294/19350959>). Could it be that the region north of Coney Island is more likely to be affected (in your study compared to CC) because you estimate higher subsidence rates there?

We sincerely thank the reviewer for this important observation. Their comment was key in identifying an error resulting from the merge of two different DEMs in New York city. The exposed areas in the previous manuscript from southern New York (Coney Island), is

a result of errors in the DEM data. This DEM data has been replaced with a single DEM file for NY city. Thus, large differences in the exposed areas in Coney Island is not as a result of the higher estimated VLM rates (compare the new figure 2 from the updated manuscript), but due to errors in the previous DEM. A comprehensive verification has been conducted on other DEMs to ensure their accuracy. Regarding the areas around the Hackensack River, it is worth noting that our focus is solely on New York City and Jersey City, and as such, those areas were not included in the analysis. We sincerely thank the reviewer for their comments.

Furthermore, as previously highlighted, to make a comparison and determine if differences in exposure are as a result of the VLM, we have to keep all other variables constant (DEM, and SLR), while varying the VLM. To that end, we have now included a comparative analysis of our study's exposure against the RSLR from the IPCC scenarios including the VLM (see figure 5). We observe that in cities where IPCC VLM rates at tide gauges are similar to the InSAR VLM within the city or where the contribution of VLM is minimal, the disparities between the estimated exposure is modest. However, the estimated exposure is underestimated in cities where the IPCC VLM underestimates the contribution of VLM. Similarly, in cities where the IPCC VLM overestimates the contribution of VLM the estimated exposure is also overestimated. This underscores the need for accurately incorporating spatially variable VLM to better prepare coastal communities for future challenges.

P6L33: Isn't it rather the spatial resolution that is unprecedented? GNSS has a much higher temporal resolution.

Yes, that is the case, updated to just spatial resolution.

P7L6-7: Could this inequality effect be even stronger, when extreme sea levels from storm surges, hurricanes etc. are taken into account (as you explain below)?

Certainly!! Also, as alluded to in the text, it is not just the hazard themselves that cause these inequalities, but failure to recover from extreme events, such as storm surges and hurricanes, tend to exacerbate the inequalities. Note that we now include the home value inequality in this section (thanks to suggestions from reviewer 1).

P7L1-3: 'approximately' reflecting the demographic distribution?

We thank the reviewer for their comments. This has been included in the text.

P8L12, Fig. 5, and sup. Fig. 9: Again, I may have misunderstood something, but to follow up on previous comments the MHW in New York is 0.66m (Excel Table 12). Does this mean that, even in the case defended here, the area exposed at high tide today is already at least as large as the projected area exposed in 2050 (red, without high tide flooding)?

This has been updated throughout the manuscript. To reflect the current (2020) exposure to high tide and the 2050 exposure are the additional exposure considering 2020 exposure as a baseline.

P8L23: '76 to 99% of the exposed area, population, and properties being situated on the Atlantic coast.' Does 99% refer to population and properties?

We thank the reviewer for their comment. In the updated manuscript, we now differentiate the percentages for area, population, and properties.

Supplementary Figures/Tables of the main File:

Fig: 2k: 'error bars represent the lower (17th percentile) and upper (83rd percentile) bounds for projected SLR'. Does that mean that the VLM uncertainty intervals, or DEM uncertainties are not included here?

In the previous manuscript that was the case. This has now been updated in the current manuscript. We now propagate all error sources including DEM and VLM errors.

Table 2: Please indicate the period of these projections, and the IPCC SLR scenario.

Supplementary Table 2 has been included in the manuscript.

Fig. 5: Maybe mention the IPCC SLR scenario in the caption.

This has been included in the figure.

Materials and Methods:

P10L42: If the original data source is NGL, I would acknowledge that, together with Shirzaei et al. 36.

This has been included on page 12; line 25.

P12: High-Tide Estimates: Does that mean you use the MSL (mean sea level) datum computed over 1983-2001? If so, to estimate exposure in 2050, wouldn't you need to compute the changes starting from that period? Or, over which period do you compute the accumulated SLC and VLM effects? Over 2020-2050 or starting from the 1983-2001 period?

Yes, we thank the reviewer for this observation. We have updated the analysis in the manuscript to represent areas exposed to high tide currently (2020) and additional areas that would be exposed by 2050 due to VLM and SLR. For this the high tide uses NAVD88 datum consistent with the DEM data.

P12L44: 'assuming a linear VLM rate to 2050', when do you start this extrapolation?

We begin this extrapolation from the base year of the DEM (2015 to 2019) to estimate the exposure by 2020 and calculate the additional exposure by 2050.

P13L11: 'We consider a census block inundated if greater than 20% of its area is inundated'. Will assuming a different figure here (20%) massively affect the results presented? Is this assumption of how much area needs to be flooded (to be counted as flooded) linearly proportional to your final estimates of the area affected? For example, if you assume that 40% has to be flooded (i.e. twice the current assumption) for a census block to be counted, would this mean that on average only half the population/area would be affected? Why is 20% a conservative estimate? Is it based on the distribution of street and building heights, etc.?

We thank the reviewer for their comments. To determine the 20% exposure threshold for individual census blocks, an empirical analysis was performed across six representative cities. Analysis of the distribution of exposed regions within these blocks shows that the percent exposed areas for these cities shows an extreme value distribution, with a

median percent exposed area within a range of 18-23% across the cities. Notably, the distribution revealed a pronounced decrease beyond the 10% mark. While there would be differences in the estimated population/properties exposure, we find that on the basis of these empirical findings, a 20% threshold is an appropriate criterion for the quantification of exposure. See supplementary Figure 16 and page 15; lines 18 to 25 of the manuscript. We now remove the reference to conservative estimate from the manuscript.

P10L10: Maybe check again if this link works? It didn't work for me.

We have checked all the links to ensure they work properly. Thanks!

Supplementary File:

Fig. 7: In the caption (of a, b, and c) it is not clarified that the period over which the changes will occur is 2020-2050 (that's only done in the main text) by assuming constant trends over this period

This has now been included in the caption for the supplementary figure.

Fig. 9. What scenario is used here?

We thank the reviewer for this observation. This has been included in the figure caption.

Reviewer Reports on the First Revision:

Referees' comments:

Referee #1 (Remarks to the Author):

I thank the authors for addressing my previous comments. I think the paper is improved, but I still have some remaining concerns.

Comparison to IPCC

Thank you for adding the comparison (lines 254-279, Figure 5) to the tide-gauge derived VLM estimates from AR6 (updated from Kopp et al 2014) and to exposure under the IPCC RSL rise projections. This is key to showing the novelty of the results here.

In doing this comparison, please do not simply separate into 'overestimate' and 'underestimate', but also into 'not distinguishable within uncertainty.' Many of the comparisons identified as 'underestimates' or 'overestimates' in fact fall within this third category.

It is also worth examining more specifically cities where the average VLM agree within uncertainty but exposure does not. Particularly worthy of discussion in this regards are where area exposure and population/property exposure differ differently -- e.g., in Miami area exposure is definitely smaller but population exposure overlaps within uncertainty. What is this saying about the spatial variability within the city?

More broadly, this discussion needs expansion where the discrepancies are large, since this is a major finding. The tide gauge-based VLM estimates differ most from the InSAR VLM estimates in the Gulf coast, in both directions (e.g., underestimate VLM at Biloxi, huge overestimate at New Orleans and Freeport). Explain why.

I would also appreciate knowing the scale of the analysis at which the differences matter. For example, in the abstract, the paper focuses on the total among the 32 cities, and elsewhere the paper looks at total totals for each of the three coastlines (e.g., Table 1). Please create a version of these results using the IPCC projections so it is possible to assess whether the differences (which, indeed, are large for some cities) are large at a regional or national scale.

Please also clarify how the average InSAR VLM rate for each city is constructed in Figure 5d.

Inequality analysis

In lines 305-317:

Please consider using the term 'minoritized' rather than 'minority' to refer to these socially constructed groupings.

The data for Asian populations seems to not align entirely with the text -- e.g., the paper states that 11.5-26.3% exposure of Asian individuals is an overrepresentation compared to a 17.8% proportion of the total population, while it appears to be a spread around the average.

In lines 319--333:

In the economic inequality analysis, the authors apply a t-test in a context with the assumptions of

this test clearly do not hold. There is not a good reason to think that property values within a city are normally distributed, and Fig. S11 gives good reason to think this is not the case. Accordingly, a non-parametric test (e.g., K-S test) would be appropriate for evaluating whether two distributions are distinguishable.

My take away from the figures is that exposed properties tend to be representative, but where they clearly differ (e.g., Biloxi, Galveston) they are overvalued.

I am also surprised by those plots showing extremely narrow distributions of property values (e.g., Galveston-TP, Foster City-EP). Please check for plotting errors, as in these cases the uncertainty range for TP and EP do not seem different enough to justify the absence of a visible bar.

Propagation of uncertainty

The authors' equation 3 appears to overestimate uncertainty by assuming perfect correlation between SLR uncertainty and the quadratic sum of DEM and VLM uncertainty. I would suggest instead

$$\text{Inun}_{\text{low}} = \text{Inun}_{\text{med}} + \sqrt{(\text{DEM}_{\text{err}})^2 + ((t-t_0)\text{VLM}_{\text{SD}})^2 + (.5*(\text{SLR}_{\{83\}} - \text{SLR}_{\{17\}})^2)}$$
 and
$$\text{Inun}_{\text{up}} = \text{Inun}_{\text{med}} - \sqrt{(\text{DEM}_{\text{err}})^2 + ((t-t_0)\text{VLM}_{\text{SD}})^2 + (.5*(\text{SLR}_{\{83\}} - \text{SLR}_{\{17\}})^2)}$$

This assumes that the SLR distributions are approximately normal (and thus the difference between the 17th and 83rd percentile projections is approximately 2 standard deviations), which should not be a terrible assumption for 2050, though would not be appropriate later in the century.

Defended scenario

It was not possible for me to figure out how the 'defended scenairo' was constructed.

Minor comments

line 17: "area IS" (not "area are")

line 33: Correct IPCC reference for historical sea-level change should be to IPCC AR6 WG1 (Fox-Kemper et al 2021) not WG2.

line 48-49: The statement that "socioeconomic losses from climate-induced sea-level rise (SLR) are likely to dominate other climate change consequences in the US" considers to be wrong and contradicts the cited source, which clearly shows the dominance of temperature-related mortality impacts.

line 62: "variation" is a confusing word choice -- you are talking about differences among emissions scenarios, but this could be interpreted as geographic or temporal variability.

line 89-95: This parenthetical note is unreadably long. Use a table.

line 181, 213, maybe elsewhere: Total or additional exposure? For a variety of reasons already discussed, focus throughout should be on additional exposure.

line 245: IPCC does not have stations. There are PSMSL tide-gauge stations included in the IPCC projections.

line 301-304: The sentence is hard to follow.

line 429-432: Sentence fragments.

line 568: I don't see a supplementary table 12 corresponding to this description.

Throughout: References to "SLR" are unclear and inconsistent about where they are talking about relative sea-level rise and geocentric sea-level rise (i.e., whether they include VLM). Please clarify.

Fig. 1: Unclear whether the SLR plots shown include VLM, and if so, from which source.

Table 1: Please add total across all 3 coast lines, in line with the abstract.

Referee #2 (Remarks to the Author):

The authors have thoroughly addressed my previous comments and, as mentioned above, provide a very comprehensive analysis of the impacts of SLR for the US coast. I think with the separation of currently exposed areas vs. projected additional exposed areas, the additional comparison with IPCC VLM values, and the incorporation of VLM errors, many of the results become much more accurate and understandable.

However, I think that the analysis presented to reinforce one of the key messages of this paper, that "not accounting for spatially variable VLM within cities may lead to inaccurate projections of expected exposure" (abstract), still requires some minor improvements as it is currently presented:

The special value of InSAR is now demonstrated by comparing the VLM estimates with the IPCC values on page 6. It turns out that on average the induced differences in exposure between InSAR and IPCC VLM are not that significant (also taking into account the VLM uncertainties), while for some individual cities they are. I think this is a valuable comparison, but two questions remain:

- Are VLM values exchanged for IPCC and InSAR? Based on Suppl. Table S11, it appears that the IPCC and InSAR VLM values shown in the x- and y-axis scatter plots (and probably in the map in Fig. 9a) are probably incorrect. This may reverse what is written in paragraphs L254-279: e.g. "current IPCC projections underestimate VLM measurements at 58% of the stations and overestimate VLM measurements at 24% of the stations (Supplementary Fig. 9)". The IPCC estimates are therefore (on average) associated with stronger subsidence than found in this study. This could also affect Fig. 5, although I think the negatively exposed area in Fig. 5a actually correctly indicates where the IPCC values cause more exposure.

- Accuracy of InSAR VLM estimates: The authors report that validation of InSAR VLM estimates with 756 GNSS trends yields a standard deviation of 1.5 mm/year. The differences between IPCC and InSAR VLM (in Table S11) have a standard deviation of 1.33 mm/yr, which is even lower than the standard deviation obtained from the comparison with the GNSS (~ground truth) values. This raises the question of how significant these differences in VLM values (IPCC vs. InSAR) and the derived differences in exposure are in this case. Are the InSAR VLMs really accurate enough to say that there is a significant difference (given the standard deviation of 1.5 mm/year vs. GNSS)? I understand that there are much fewer samples (IPCC vs. InSAR, n=33) compared to the GNSS station comparison (756), and that even GNSS and IPCC values are not perfectly accurate. However, I really wonder how confident the authors are about the significance of the difference between the InSAR and IPCC VLM values. Aren't the formal uncertainties (i.e. the median uncertainties of the InSAR std in Table S11 of 0.32 mm/yr) also too low to reflect the true accuracy of the InSAR estimates? This may also affect the significance of the differences (or error

bars) in exposure when using IPCC vs. InSAR (Fig. 5). Therefore, I think it might be worth mentioning the imperfect accuracy of the InSAR data somewhere in the main text. Maybe in paragraphs 254-279, also because in Fig. 5 the uncertainties of the exposure differences are already quite large and probably still too optimistic.

Apart from these issues, I fully agree with the authors on the need to have this spatial resolution of InSAR (as opposed to the sparse IPCC TG or GNSS estimates). However, the problem that they are sometimes not very accurate and that the formal errors may not represent this accuracy very well should be mentioned. In general, I think that since the authors really put much more effort into comparing their results with other VLM estimates, they illuminate the impacts and differences in SL changes from many different perspectives, which will be very useful for the SL community for further developments.

Minor comments:

I have noticed that the paper is relatively long, of course also due to the responses to the reviews. Although it's probably up to the editor to decide how long it can be, I'm afraid that some of these analyses in the main text will have to be moved later.

L16: Compared to the previous version of the article, the exposed land area (1000-1,389km²) approximately stays the same, while number of people/properties is drastically reduced. Could there be an error here? I know there are changes due to the treatment of currently vs. additionally exposed land, but how can this be explained?

Paragraph from L254 and related to the previous comment:

It took me some time to understand to what the terms over/underestimated are referred to. Maybe add that somewhere in the beginning that it refers to IPCC (L259? and in Fig. 5).

Fig. 5: A question that arises for me is, that even though there are local differences between the IPCC and InSAR, the error bars are often so large that the differences in the exposed areas are often not significant. Maybe that should be said somewhere.

Fig. 9. (Supp. File):

Does table S11 represent the values shown in this Figure?

If yes, I think there are some inconsistencies here. the IPCC or InSAR VLM values are mixed up, either in the table or in the scatter plots. It seems it's mixed up in the scatter plot because the projections in f,g, and h match with the table. Does that mean InSAR underestimates IPCC estimates (in terms of subsidence)?

In h the InSAR VLM SLC should be 0.017 cm/yr not 0.17 cm/yr

L175 (Supp. File): The 10% error buffer seems arbitrary, right? I understand that in this case it's difficult to find an objective threshold to decide the number of over/underestimated cases, and I agree that some decision has to be made. However, and unfortunately, this threshold really has a strong influence on the number of over/underestimated cases. Do you think there would be big differences if you used the combined uncertainties of the IPCC and InSAR estimates, or even the 1.5 mm/year stdev as a threshold (based on the InSAR vs. GNSS differences)?

L174 (Supp. File): Brackets should in the denominator should probably be around (IPCC VLM + InSAR VLM)/2

If the IPCC and InSAR values are indeed mistaken in Fig. 9. potentially the statements in L272-273 (and L259-260?) should be corrected.

Author Rebuttals to First Revision:

Reviewer #1 (Remarks to the Author): (Reviewers comments in normal text, response to reviewers are in bold)

I thank the authors for addressing my previous comments. I think the paper is improved, but I still have some remaining concerns.

Comparison to IPCC

Thank you for adding the comparison (lines 254-279, Figure 5) to the tide-gauge derived VLM estimates from AR6 (updated from Kopp et al 2014) and to exposure under the IPCC RSL rise projections. This is key to showing the novelty of the results here.

We thank the reviewer for their current and past reviews, their suggestions and comments were key to updating and improving the manuscript.

In doing this comparison, please do not simply separate into 'overestimate' and 'underestimate', but also into 'not distinguishable within uncertainty.' Many of the comparisons identified as 'underestimates' or 'overestimates' in fact fall within this third category.

This is indeed the case. We have added this category when discussing these differences on lines 263 - 267. We thank the reviewer for this suggestion.

It is also worth examining more specifically cities where the average VLM agree within uncertainty but exposure does not. Particularly worthy of discussion in this regards are where area exposure and population/property exposure differ differently -- e.g., in Miami area exposure is definitely smaller but population exposure overlaps within uncertainty. What is this saying about the spatial variability within the city? More broadly, this discussion needs expansion where the discrepancies are large, since this is a major finding. The tide gauge-based VLM estimates differ most from the InSAR VLM estimates in the Gulf coast, in both directions (e.g., underestimate VLM at Biloxi, huge overestimate at New Orleans and Freeport). Explain why.

We have expanded the discussion in this section on lines 277 – 285 and supplementary Figure 9 to include the comparison for some specific cities, including Miami, Biloxi, and New Orleans. The differences in the estimated area exposure for cities on the U.S. East Coast (e.g. Charleston, Savannah, and Miami) exist because the overestimated/underestimated areas occur primarily in the non-urban areas/wetlands. For the U.S. Gulf Coast, these differences are more widespread in both urban/non-urban areas, resulting in differences in all exposure categories.

To discuss the differences in the VLM, we included an additional comparison of VLM at the location of tide gauges across the U.S. (supplementary Figure 11). Note that this is different from the data presented in Figure 5d. In Fig. 5d we compare VLM at the tide gauge station (from IPCC VLM) versus the average VLM within the entire city (from InSAR VLM). For comparison of VLM at the location of the tide gauges (supplementary Figure 11), we average InSAR VLM within a 200m radius of each station. The comparison shows 59% consistency between VLM at the tide gauge stations. Even in cities where VLM measurements at the tide gauge location are similar from both datasets, significant differences also exist between what happens within the city itself and at the tide gauge station. An explanation for these differences is provided in lines 293 - 298 of the

manuscript. “Tide gauge stations situated at the peripheries of urban areas may not accurately capture the contemporary dynamics of spatially variable VLM within the cities themselves.” Thus, such differences are expected. This is true in the case of Biloxi and New Orleans, where although we find consistencies between InSAR and IPCC VLM at the location of the tide gauge (supplementary Fig. 11), there are differences between the VLM at the tide gauge and within the cities. Also, in San Diego, the average VLM in the city is lower than the VLM measured at the tide gauge station.

I would also appreciate knowing the scale of the analysis at which the differences matter. For example, in the abstract, the paper focuses on the total among the 32 cities, and elsewhere the paper looks at total totals for each of the three coastlines (e.g., Table 1). Please create a version of these results using the IPCC projections so it is possible to assess whether the differences (which, indeed, are large for some cities) are large at a regional or national scale.

We have included table 2 for the IPCC RSLR exposure, which is constructed in the same manner as table 1. A discussion about this has also been included in lines 264 – 265 of the manuscript.

Please also clarify how the average InSAR VLM rate for each city is constructed in Figure 5d.

This has been clarified in the figure caption. The average VLM is similar to what is shown in Fig. 1e, this is the average spatially variable VLM used in the inundation analysis for this study.

Inequality analysis

In lines 305-317:

Please consider using the term 'minoritized' rather than 'minority' to refer to these socially constructed groupings.

Minority has been replaced with minoritized in the inequality analysis section of the manuscript.

The data for Asian populations seems to not align entirely with the text -- e.g., the paper states that 11.5-26.3% exposure of Asian individuals is an overrepresentation compared to a 17.8% proportion of the total population, while it appears to be a spread around the average.

11.5 - 26.3% refers to the range for the lower and upper bounds. We have now specified this specifically using the median and upper bounds in the manuscript.

In lines 319--333:

In the economic inequality analysis, the authors apply a t-test in a context with the assumptions of this test clearly do not hold. There is not a good reason to think that property values within a city are normally distributed, and Fig. S11 gives good reason to think this is not the case. Accordingly, a non-parametric test (e.g., K-S test) would be appropriate for evaluating whether two distributions are distinguishable.

We have repeated the analysis using a Kolmogorov-Smirnov (K-S) test. While we generally find the same general conclusions as the t-test. It is important to highlight that

the percentage of rejection are significantly higher compared to the t-test. This is because unlike parametric tests (such as the t-test) where we compare the parameters (mean and variances), the K-S test is sensitive to the behavior of the tails, i.e., extreme values. If the distributions have the same mean, but very different behavior at the tails, the K-S test will often say they are different. For the inequality analysis, the mean and variances should matter. Nevertheless, we retain the K-S test in the manuscript per the reviewer's suggestion.

My take away from the figures is that exposed properties tend to be representative, but where they clearly differ (e.g., Biloxi, Galveston) they are overvalued.

While a qualitative analysis of the boxplot figure may lead to such conclusions. However, the quantitative analysis (K-S test) is the true test for qualifying such differences. We acknowledge that the other cities where the exposed properties are overvalued were worth mentioning. This has been included in the manuscript on line 348.

I am also surprised by those plots showing extremely narrow distributions of property values (e.g., Galveston-TP, Foster City-EP). Please check for plotting errors, as in these cases the uncertainty range for TP and EP do not seem different enough to justify the absence of a visible bar.

We have checked the property values and there are no plotting errors. The narrow range for uncertainty for TP in Galveston (not including Texas City) exists because the property value are estimated from 3 zip codes present within the city (see figure 1 below). Exposed properties in Foster city are also estimated from only 3 zip codes, causing a narrow margin of uncertainty.

Figure 1: Home value for zip-codes in Galveston.

Propagation of uncertainty

The authors' equation 3 appears to overestimate uncertainty by assuming perfect correlation between SLR uncertainty and the quadratic sum of DEM and VLM uncertainty. I would suggest instead

$$\text{Inun}_{\text{low}} = \text{Inun}_{\text{med}} + \sqrt{(\text{DEM}_{\text{err}})^2 + ((t-t_0)\text{VLM_SD})^2 + (.5*(\text{SLR}_{\text{83}} - \text{SLR}_{\text{17}})^2)}$$
 and

$$\text{Inun}_{\text{up}} = \text{Inun}_{\text{med}} - \sqrt{(\text{DEM}_{\text{err}})^2 + ((t-t_0)\text{VLM_SD})^2 + (.5*(\text{SLR}_{\text{83}} - \text{SLR}_{\text{17}})^2)}$$

This assumes that the SLR distributions are approximately normal (and thus the difference between the 17th and 83rd percentile projections is approximately 2 standard deviations), which should not be a terrible assumption for 2050, though would not be appropriate later in the century.

We thank the reviewer for this observation. We in fact, implemented the error propagation in the manner described above by the reviewer, which is obviously the correct way of doing it. However, the reported equation in the manuscript was an attempt to simplify the equation, which was incorrect. We have now updated equation (3) in the manuscript.

Defended scenario

It was not possible for me to figure out how the 'defended scenairo' was constructed.

We thank the reviewer for this observation. This has been included on lines 406 – 408.

Minor comments

line 17: "area IS" (not "area are")

We thank the reviewer for this observation. This has been corrected.

line 33: Correct IPCC reference for historical sea-level change should be to IPCC AR6 WG1 (Fox-Kemper et al 2021) not WG2.

This has been updated.

line 48-49: The statement that "socioeconomic losses from climate-induced sea-level rise (SLR) are likely to dominate other climate change consequences in the US" considers to be wrong and contradicts the cited source, which clearly shows the dominance of temperature-related mortality impacts.

We thank the reviewer for this comment. This has been updated in the manuscript.

line 62: "variation" is a confusing word choice -- you are talking about differences among emissions scenarios, but this could be interpreted as geographic or temporal variability.

Variation has been replaced with differential.

line 89-95: This parenthetical note is unreadably long. Use a table.

Modified to a bullet for each coast.

line 181, 213, maybe elsewhere: Total or additional exposure? For a variety of reasons already discussed, focus throughout should be on additional exposure.

This has been corrected throughout the manuscript. We thank the reviewer for this observation.

line 245: IPCC does not have stations. There are PSMSL tide-gauge stations included in the IPCC projections.

This has been updated. We thank the reviewer for this observation.

line 301-304: The sentence is hard to follow.

The sentence has been rewritten for clarity.

line 429-432: Sentence fragments.

This sentence has been rewritten for clarity. We thank the reviewer for their observation.

line 568: I don't see a supplementary table 12 corresponding to this description.

We thank the reviewer for this observation. This has been updated to Supplementary table 23.

Throughout: References to "SLR" are unclear and inconsistent about where they are talking about relative sea-level rise and geocentric sea-level rise (i.e., whether they include VLM). Please clarify.

Throughout the manuscript, we refer to relative sea level rise, which includes VLM as "relative SLR" and sea level rise without the contribution of VLM as simply SLR. This has been included in the introduction on lines 37 – 39.

Fig. 1: Unclear whether the SLR plots shown include VLM, and if so, from which source.

This does not contain VLM as indicated from the comments above.

Table 1: Please add total across all 3 coast lines, in line with the abstract.

This has been included in the table. Please note that the abstract references the defended scenario and not the undefended scenario shown in table 1.

We sincerely appreciate the reviewer's helpful comments and suggestions on our manuscript.

Reviewer #2 (Remarks to the Author): (Reviewers comments in normal text, response to reviewers are in bold)

The authors have thoroughly addressed my previous comments and, as mentioned above, provide a very comprehensive analysis of the impacts of SLR for the US coast. I think with the separation of currently exposed areas vs. projected additional exposed areas, the additional comparison with IPCC VLM values, and the incorporation of VLM errors, many of the results become much more accurate and understandable.

We appreciate the reviewer's insightful feedback, which have been really helpful to improve this manuscript. We have incorporated the suggestions and comments in this round of review into the manuscript as discussed below.

However, I think that the analysis presented to reinforce one of the key messages of this paper, that "not accounting for spatially variable VLM within cities may lead to inaccurate projections of expected exposure" (abstract), still requires some minor improvements as it is currently presented:

The special value of InSAR is now demonstrated by comparing the VLM estimates with the IPCC values on page 6. It turns out that on average the induced differences in exposure between InSAR and IPCC VLM are not that significant (also taking into account the VLM uncertainties), while for some individual cities they are. I think this is a valuable comparison, but two questions remain:

- Are VLM values exchanged for IPCC and InSAR? Based on Suppl. Table S11, it appears that the IPCC and InSAR VLM values shown in the x- and y-axis scatter plots (and probably in the map in Fig. 9a) are probably incorrect. This may reverse what is written in paragraphs L254-279: e.g. "current IPCC projections underestimate VLM measurements at 58% of the stations and overestimate VLM measurements at 24% of the stations (Supplementary Fig. 9)". The IPCC estimates are therefore (on average) associated with stronger subsidence than found in this study. This could also affect Fig. 5, although I think the negatively exposed area in Fig. 5a actually correctly indicates where the IPCC values cause more exposure.

We thank the reviewer for this observation. The VLM values are not incorrect, but the supplementary Figure 9 (now supplementary figure 10) and Figure 5d refers to two different analysis, which we have now clarified in the manuscript. For Figure 5d, we compare the average InSAR VLM "within" the cities (utilized in our exposure analysis), with the VLM at the tide gauge stations incorporated into the IPCC RSLR projections (utilized in the IPCC-derived exposure analysis). For supplementary Figure 10, we compare the VLM at the tide gauge stations incorporated into the IPCC RSLR projections, with InSAR VLM within 200m radius of the tide gauge stations. This analysis was important to highlight that regardless of the consistency in VLM measurements at the tide gauge stations, these stations, often situated at the edge of the cities, may often have differing rates when compared to the cities themselves. We have highlighted this point in the manuscript. We have also included the dataset for supplementary Figure 10 (previously missing) in supplementary table 12. Note that this figure has also been updated to include 74 stations. We apologize for the lack of clarity.

- Accuracy of InSAR VLM estimates: The authors report that validation of InSAR VLM estimates with 756 GNSS trends yields a standard deviation of 1.5 mm/year. The differences between

IPCC and InSAR VLM (in Table S11) have a standard deviation of 1.33 mm/yr, which is even lower than the standard deviation obtained from the comparison with the GNSS (~ground truth) values. This raises the question of how significant these differences in VLM values (IPCC vs. InSAR) and the derived differences in exposure are in this case. Are the InSAR VLMs really accurate enough to say that there is a significant difference (given the standard deviation of 1.5 mm/year vs. GNSS)? I understand that there are much fewer samples (IPCC vs. InSAR, n=33) compared to the GNSS station comparison (756), and that even GNSS and IPCC values are not perfectly accurate. However, I really wonder how confident the authors are about the significance of the difference between the InSAR and IPCC VLM values. Aren't the formal uncertainties (i.e. the median uncertainties of the InSAR std in Table S11 of 0.32 mm/yr) also too low to reflect the true accuracy of the InSAR estimates? This may also affect the significance of the differences (or error bars) in exposure when using IPCC vs. InSAR (Fig. 5). Therefore, I think it might be worth mentioning the imperfect accuracy of the InSAR data somewhere in the main text. Maybe in paragraphs 254-279, also because in Fig. 5 the uncertainties of the exposure differences are already quite large and probably still too optimistic.

We thank the reviewer for this observation. This is a really important point! We cannot statistically compare the differences between VLM rates in Figure 5d from the InSAR and IPCC VLM, because they are obtained at different locations. We can, however, do a statistical comparison for VLM rates obtained within 200 m radius of the tide gauge used by IPCC as shown in supplementary figure 10. This has been included in the analysis shown in supplementary table 12.

The issue of error underestimation when applying least squares to a large set of observations with few unknowns (i.e., a large degree of freedom) is a well-known problem that arises in most optimization exercises. That is because the optimum unknown value's standard deviation is proportional to each observation's uncertainty divided by the square root of the degree of freedom.

We have included a few sentences about the imperfect accuracy of InSAR data in the main text in lines 298 – 303.

Apart from these issues, I fully agree with the authors on the need to have this spatial resolution of InSAR (as opposed to the sparse IPCC TG or GNSS estimates). However, the problem that they are sometimes not very accurate and that the formal errors may not represent this accuracy very well should be mentioned. In general, I think that since the authors really put much more effort into comparing their results with other VLM estimates, they illuminate the impacts and differences in SL changes from many different perspectives, which will be very useful for the SL community for further developments.

We thank the reviewer for their comments! We also appreciate and acknowledge their suggestion to make that comparison. The current manuscript greatly benefitted from their excellent reviews.

Minor comments:

I have noticed that the paper is relatively long, of course also due to the responses to the reviews. Although it's probably up to the editor to decide how long it can be, I'm afraid that some of these analyses in the main text will have to be moved later.

Yes, we notice that as well and we have included even more analyses in this current review. We hope we can move some of the analysis and discussion to the supplement after the review process.

L16: Compared to the previous version of the article, the exposed land area (1000-1,389km²) approximately stays the same, while number of people/properties is drastically reduced. Could there be an error here? I know there are changes due to the treatment of currently vs. additionally exposed land, but how can this be explained?

Yes, that is the case. We are grateful for the reviewer's keen observations, which led us to identify and rectify the error in the DEM for New York. In our previous analysis, New York had an area exposure of 100 km² and a population exposure of 800,000 individuals. Upon correction of the DEM, we observed a substantial reduction in the population exposure estimates for New York to about 3,000. This adjustment has pronounced impacted the overall population and properties exposure while not having a huge impact on the area exposure.

Paragraph from L254 and related to the previous comment:

It took me some time to understand to what the terms over/underestimated are referred to. Maybe add that somewhere in the beginning that it refers to IPCC (L259? and in Fig. 5).

We have modified the sentences in this paragraph and supplementary Figure 5 to show that it refers to IPCC.

Fig. 5: A question that arises for me is, that even though there are local differences between the IPCC and InSAR, the error bars are often so large that the differences in the exposed areas are often not significant. Maybe that should be said somewhere.

Yes, that is the case, we have now included a new category to indicate 'not distinguishable within uncertainty.'

Fig. 9. (Supp. File):

Does table S11 represent the values shown in this Figure?

If yes, I think there are some inconsistencies here. the IPCC or InSAR VLM values are mixed up, either in the table or in the scatter plots. It seems it's mixed up in the scatter plot because the projections in f,g, and h match with the table. Does that mean InSAR underestimates IPCC estimates (in terms of subsidence)?

No, it does not. We have now included supplementary table S12 which represents the data for supplementary Fig. 9 (now supplementary fig. 10). We now do this analysis for 74 stations. We have included additional note to the figure caption for clarity.

In h the InSAR VLM SLC should be 0.017 cm/yr not 0.17 cm/yr

San diego is a good example of stations were the VLM at the tide gauge is not representative of the VLM within the city themselves. The average InSAR VLM within 200m of the tide gauge station is -1.8 mm/year. However, the average VLM within the city is -0.17mm/year. This is not an error. Fig. 5d and supplementary Fig. 10 are different.

L175 (Supp. File): The 10% error buffer seems arbitrary, right? I understand that in this case it's difficult to find an objective threshold to decide the number of over/underestimated cases, and I agree that some decision has to be made. However, and unfortunately, this threshold really has a strong influence on the number of over/underestimated cases. Do you think there would be big differences if you used the combined uncertainties of the IPCC and InSAR estimates, or even the 1.5 mm/year stdev as a threshold (based on the InSAR vs. GNSS differences)?

L174 (Supp. File): Brackets should in the denominator should probably be around $(\text{IPCC VLM} + \text{InSAR VLM})/2$

Yes! The 10% error buffer is arbitrary. We cannot utilize a 1.5 mm/year buffer range for each tide gauge station. This would assume a systematic error in the InSAR data as opposed to a random error. In the case of the InSAR-GNSS comparison, the 1.5 m/year standard deviation was for all stations, not each individual station. We have opted for a z-test to statistically test for differences between the InSAR and IPCC VLM at the tide gauge stations. This is useful because such tests rely on both the mean values and their standard deviation. This has been updated in the text and supplementary figures.

If the IPCC and InSAR values are indeed mistaken in Fig. 9. Potentially the statements in L272-273 (and L259-260?) should be corrected.

Yes, that would be the case. This has been clarified in the previous comments. We sincerely appreciate your reviews.

Reviewer Reports on the Second Revision:

Referees' comments:

Referee #1 (Remarks to the Author):

I think the authors for their response. I believe they have adequately addressed the bulk of my concerns, though I do have one minor concern remaining regarding terminology.

The authors write "In this study, we refer to sea-level rise that incorporates the effects of Vertical Land Motion (VLM) as 'relative SLR,' whereas 'SLR' refers to sea-level rise without VLM." This is inconsistent with standard usage in the sea-level research community, as set out in Gregory et al. (2019, 10.1007/s10712-019-09525-z) and employed, for example, by the IPCC. In standard usage, "sea-level rise" unqualified can refer to *either* geocentric or relative sea-level rise, whereas the authors appear to use it to refer to what common terminology would describe as "geocentric sea-level rise." Per Gregory et al. 2019:

"The phrases "sea-level change" (SLC) and "sea-level rise" (SLR) are often used in the literature. These make sense when referring to the phenomenon in general, but more specific terms such as relative sea-level change and global-mean sea-level rise should be preferred where relevant."

"N14 Geocentric sea-level change $\Delta\eta$: The change in local mean sea level with respect to the terrestrial reference frame."

"N15 Relative sea-level change (RSLC) ΔR : The change in local mean sea level relative to the local solid surface, i.e., the sea floor. Relative sea-level change is also called "relative sea-level rise" (RSLR). (See Sect. 6 for an exposition of the relationship of RSLC to other quantities.)"

The use of "sea-level rise" generically to refer to "geocentric sea-level rise" in particular contradicts common use in coastal risk management, since it is relative sea-level rise, not geocentric sea-level rise, that creates risk.

I strongly urge the authors to adopt standard terminology based on Gregory et al. (2019).

I also have one minor suggestions regarding the added Table 2. The primary purpose of this table is comparison to Table 1, which at the moment requires extensive flipping back and forth. It might be worth considering merging these into a single Table that would allow for more direct comparison.

Referee #2 (Remarks to the Author):

Dear Author,

All my previous comments have been clarified and I don't have any further comments (except for adding 'Table' in the supplementary file Line 187 after 'supplementary'). This is a really comprehensive paper and an important contribution to a better understanding of the consequences of sea level rise and VLM!

Author Rebuttals to Second Revision:

Reviewer #1 (Remarks to the Author): (Reviewers comments in normal text, response to reviewers are in bold)

I think the authors for their response. I believe they have adequately addressed the bulk of my concerns, though I do have one minor concern remaining regarding terminology.

The authors write "In this study, we refer to sea-level rise that incorporates the effects of Vertical Land Motion (VLM) as 'relative SLR,' whereas 'SLR' refers to sea-level rise without VLM." This is inconsistent with standard usage in the sea-level research community, as set out in Gregory et al. (2019, 10.1007/s10712-019-09525-z) and employed, for example, by the IPCC. In standard usage, "sea-level rise" unqualified can refer to *either* geocentric or relative sea-level rise, whereas the authors appear to use it to refer to what common terminology would describe as "geocentric sea-level rise." Per Gregory et al. 2019:

"The phrases "sea-level change" (SLC) and "sea-level rise" (SLR) are often used in the literature. These make sense when referring to the phenomenon in general, but more specific terms such as relative sea-level change and global-mean sea-level rise should be preferred where relevant."

"N14 Geocentric sea-level change $\Delta\eta$: The change in local mean sea level with respect to the terrestrial reference frame."

"N15 Relative sea-level change (RSLC) ΔR : The change in local mean sea level relative to the local solid surface, i.e., the sea floor. Relative sea-level change is also called "relative sea-level rise" (RSLR). (See Sect. 6 for an exposition of the relationship of RSLC to other quantities.)"

The use of "sea-level rise" generically to refer to "geocentric sea-level rise" in particular contradicts common use in coastal risk management, since it is relative sea-level rise, not geocentric sea-level rise, that creates risk.

I strongly urge the authors to adopt standard terminology based on Gregory et al. (2019).

We sincerely thank the reviewer for their comments. We now adopt the terminology by Gregory et al. (2019): geocentric sea level rise (SLR) for SLR without vertical land motion (VLM) and relative SLR for SLR incorporating VLM.

I also have one minor suggestions regarding the added Table 2. The primary purpose of this table is comparison to Table 1, which at the moment requires extensive flipping back and forth. It might be worth considering merging these into a single Table that would allow for more direct comparison.

We thank the reviewer for this suggestion. We have merged table 1 and 2 into a single table.

Reviewer #2 (Remarks to the Author): (Reviewers comments in normal text, response to reviewers are in bold)

Dear Author,

All my previous comments have been clarified and I don't have any further comments (except for adding 'Table' in the supplementary file Line 187 after 'supplementary'). This is a really comprehensive paper and an important contribution to a better understanding of the consequences of sea level rise and VLM!

We sincerely thank the reviewer for their comprehensive reviews. Their insights, comments, and questions were invaluable in helping us enhance this research and manuscript. The correction has been included in line 187.